# Decision-focused Sparse Tangent Portfolio Optimization

**Haeun Jeon** [* 1]   **Seunghoon Choi** [* 1]   **Hyunglip Bae** [2]   **Yongjae Lee** [3]   **Woo Chang Kim** [1]

## Abstract

Sparse tangent portfolio optimization aims to learn an interpretable, low-cardinality portfolio in the tangency direction of the mean-variance frontier. However, the associated cardinality-constrained formulation is NP-hard, and standard predict-then-optimize pipelines often misalign forecasting accuracy with downstream portfolio quality. We propose an end-to-end decision-focused learning framework that reformulates Sharpe ratio maximization as a Disciplined Parametrized Programming (DPP)-compliant convex programming layer and replaces discrete selection with a smooth top-$k$ operator enforcing an exact cardinality $k$. This enables gradient flow through prediction, asset selection, and re-optimization, allowing the predictive model to directly optimize portfolio performance. Across four major equity markets, our method achieves competitive and often superior out-of-sample Sharpe ratios compared with historical and prediction-focused baselines, with particularly strong gains in larger asset universes. Our code is publicly available.

## 1. Introduction

Originating from Markowitz's mean-variance model (Markowitz, 1952), modern portfolio theory views investment decisions through the lens of balancing risk and return at the portfolio level, rather than evaluating assets in isolation (Kim et al., 2021). By combining assets with heterogeneous risk-return characteristics, investors can construct diversified portfolios that deliver lower variance for a given level of expected return. This principle is formalized through the efficient frontier and performance measures such as the Sharpe ratio, which captures the trade-off between excess return and portfolio volatility (Sharpe, 1966). Although diversification is appealing in theory, holding too many assets is often impractical, as it raises monitoring and transaction costs and makes the portfolio harder to interpret (Woodside-Oriakhi, 2011).

For these practical reasons, portfolio managers often favor sparse allocations that concentrate exposure on a limited number of assets. A standard way to formalize this preference is through a cardinality constraint, which limits the number of nonzero positions. Enforcing such sparsity, however, introduces a fundamental tension: while restricting the number of holdings improves interpretability and reduces transaction costs, it turns portfolio construction into a challenging combinatorial problem. As a result, existing approaches must balance solution quality, computational efficiency, and flexibility under varying cardinality levels. Recently, heuristic pipelines that decouple asset selection from re-optimization have shown promise as practical tools for constructing sparse tangent portfolios, offering a favorable balance between solution quality and scalability. However, these pipelines still rely on upstream forecasts produced by a separate predictive model, and their effectiveness ultimately depends on the end-to-end interaction between prediction, discrete selection, and continuous re-optimization.

Even with strong heuristics for sparse optimization, a persistent disconnect remains between prediction and decision. Conventional frameworks typically fit models on historical data and then deploy the resulting rules to make future decisions (Fabozzi et al., 2007), but they often fail to adapt to non-stationary, fast-moving markets and shifting constraints (Schmitt et al., 2013; Hamilton, 2020; Hwang et al., 2025). To mitigate this, practitioners train machine-learning models to predict future returns and let an optimizer consume those predictions to construct portfolios. However, these predictors are usually trained under statistical losses, which are misaligned with the ultimate objective of portfolio optimization and do not account for the selection and re-optimization steps that drive realized performance. This predict-then-optimize mismatch is especially harmful under tight cardinality, where small forecast errors can cascade

---

[*]Equal contribution  [1]Department of Industrial and Systems Engineering, Korea Advanced Institute of Science and Technology, Daejeon, Republic of Korea [2]Department of Big Data Convergence, and Bigdata Research Lab, Chonnam National University, Gwangju, Republic of Korea [3]Department of Industrial Engineering, Ulsan National Institute of Science and Technology, Ulsan, Republic of Korea. Correspondence to: Yongjae Lee <yongjaelee@unist.ac.kr>, Woo Chang Kim <wkim@kaist.ac.kr>.

*Proceedings of the 43$^{rd}$ International Conference on Machine Learning*, Seoul, South Korea. PMLR 306, 2026. Copyright 2026 by the author(s).

through the discrete selection stage and magnify into substantial utility losses.

In this paper, we show that decision-focused learning (DFL) can be effectively applied to portfolio optimization with explicit cardinality constraints (Mandi et al., 2024). Unlike conventional historical estimation or predict-then-optimize pipelines, DFL trains predictive models by directly optimizing downstream decision quality rather than intermediate forecasting accuracy. We develop an end-to-end decision-focused framework for sparse portfolio optimization that enables gradient-based learning through asset selection and portfolio re-optimization. The proposed approach aligns the learning objective with risk-adjusted portfolio performance under a fixed sparsity budget, allowing the predictive model to internalize the structure of the constrained optimization problem.

Our main contributions are threefold. First, to the best of our knowledge, this is the first work to apply DFL to sparse tangent portfolio optimization with explicit cardinality constraints, bridging two research directions that have traditionally evolved separately. Second, we propose a differentiable decision layer that combines a DPP-compliant convex reformulation of Sharpe ratio maximization with a smooth top-$k$ asset selection operator, enabling end-to-end gradient flow through prediction, asset selection, and re-optimization. Finally, we conduct experiments across four markets with different baselines: Historic, Prediction-focused Learning (PFL), and DFL pipelines, showing that DFL achieves competitive or superior performance, with more gains in larger asset universes.

## 2. Related Works

### 2.1. Portfolio Optimization

The portfolio optimization problem yields the optimal portfolio weights, which specify how capital is allocated across assets. Markowitz's framework established portfolio construction as an optimization problem, and subsequent work expanded it in many directions: improved estimation of returns and covariances, alternative risk measures, and practical constraints that reflect portfolio management (Kim et al., 2024). In practice, holding many assets is costly and hard to manage, so managers often prefer sparse allocations that concentrate exposure on a small number of positions (Woodside-Oriakhi, 2011). A standard way to formalize this preference is through a cardinality constraint, which explicitly caps the number of nonzero weights.

While this constraint is conceptually straightforward, it fundamentally changes the computational nature of the problem: incorporating asset-selection discreteness into objectives turns an otherwise smooth continuous optimization into a combinatorial search, which is NP-hard in general

and therefore difficult to solve exactly for large asset universes (Garey & Johnson, 2002). Controlling portfolio cardinality has been studied extensively and can be broadly categorized into three methodological directions. One line induces sparsity indirectly through regularization or norm-based penalties, trading exact asset selection for scalability (Brodie et al., 2009; Chen & Zhu, 2014; Kremer et al., 2020; DeMiguel et al., 2009). Another line tackles the discrete constraint more directly via exact or relaxation-based algorithms such as branch-and-cut and related formulations (Bienstock, 1996; Li et al., 2006; Gao & Li, 2013; Kim & Lee, 2016; Lee et al., 2020). A third line relies on heuristic methods that trade optimality guarantees for computational efficiency in large asset universes (Chang et al., 2000; Maringer & Kellerer, 2003).

Beyond mean-variance, portfolio optimization has been studied under a variety of objectives, among which risk-adjusted performance metrics naturally motivate Sharpe ratio maximization as a single-objective formulation. Several works study Sharpe ratio maximization under explicit sparsity requirements (Sharpe, 1966). The Sharpe ratio is a standard risk-adjusted performance metric that normalizes expected return by volatility, enabling comparisons across portfolios with different risk levels. Representative approaches include SDP-based relaxations that yield tractable convex approximations of the sparse Sharpe problem (Kim et al., 2016), as well as methods that reformulate the fractional objective and apply efficient first-order algorithms with theoretical guarantees (Lin et al., 2024). Complementing these optimization-centric approaches, OSCAR (Optimize, Select with Cholesky, and Re-optimize) is a practical heuristic pipeline that alternates between continuous optimization and discrete asset selection to obtain scalable, high-quality sparse portfolios in large universes (Bae et al., 2025).

### 2.2. Decision-focused Learning

Traditional workflows that perform optimization after learning typically follow a PFL paradigm, often referred to as predict-then-optimize or two-stage learning (Lee et al., 2024). A predictive model is first trained to output point estimates of unknown problem parameters using standard statistical losses like mean squared error or cross-entropy. These predictions are then treated as fixed inputs to a downstream constrained optimization model, which computes the final decision. However, PFL does not account for the structure of the downstream decision problem. It implicitly assumes that minimizing prediction loss will lead to high-quality decisions, an assumption that often fails in practice. Therefore, models trained purely on prediction loss may perform poorly in terms of the true task objective even when their forecasts are statistically accurate.

DFL was introduced to address this mismatch by aligning

model training directly with decision quality rather than with intermediate prediction accuracy. In DFL, the prediction model and the optimization layer are integrated into a single end-to-end pipeline, and training minimizes a task loss that depends on the realized decision instead of a pure prediction loss (Mandi et al., 2024). This paradigm has been applied to diverse tasks including network design, resource allocation, routing, portfolio optimization, and goal-based investing, and typically yields substantial gains over PFL when decision quality is sensitive to structured errors in the predictions.

A central technical challenge in DFL is computing gradients of the task loss with respect to the prediction model parameters. The main difficulty lies in differentiating through the downstream optimization. One must characterize how the optimizer's solution changes in response to perturbations in the predicted cost or parameters, even though the solution map is defined only implicitly as the argmin of an optimization problem and typically has no closed-form expression, leading to zero or undefined gradients over large regions of the parameter space. Accordingly, prior DFL work typically obtains usable learning signals either by analytically differentiating the optimizer's optimality conditions when the solution map is smooth (Shah et al., 2022), or by regularizing the optimization to create stable gradients (Brodie et al., 2009; Chen & Zhu, 2014).

## 3. Preliminaries

### 3.1. Cardinality-constrained Portfolio Optimization

A standard construction of portfolio optimization for maximizing the Sharpe ratio (Sharpe, 1966) can be written as:

$$\max_{w \in \mathbb{R}^n} \quad \frac{\mu^\top w}{\sqrt{w^\top \Sigma w}} \quad \text{s.t.} \quad \mathbf{1}^\top w = 1 \qquad (1)$$

where $w \in \mathbb{R}^n$ denotes the portfolio weight vector, $\mu \in \mathbb{R}^n$ denotes the expected excess return vector, $\Sigma \in \mathbb{R}^{n \times n}$ is the covariance matrix of asset returns, and $\mathbf{1}$ is the all-ones vector. Throughout this work, we allow short selling so negative entries in $w$ are permitted. The numerator represents the portfolio's expected return, while the denominator corresponds to its risk induced by $\Sigma$.

To enhance practical interpretability and to reduce transaction costs, one often restricts the number of assets held in the portfolio via a cardinality constraint. This leads to the sparse Sharpe ratio maximization problem:

$$\max_{w \in \mathbb{R}^n} \quad \frac{\mu^\top w}{\sqrt{w^\top \Sigma w}}$$
$$\text{s.t.} \quad \mathbf{1}^\top w = 1 \qquad (2)$$
$$\text{Card}(w) \leq k$$

where $\text{Card}(w)$ counts the number of nonzero compo-

nents of $w$ with sparsity budget $k$. As discussed in Section 2.1, adding this constraint makes the problem combinatorial rather than smoothly continuous. Since cardinality-constrained portfolio selection is NP-hard in general, exact methods are typically impractical for large asset universes.

Recent work proposes OSCAR (Bae et al., 2025) as a fast heuristic for sparse tangent portfolio optimization in the classical mean-variance framework. It avoids the combinatorial complexity of cardinality-constrained portfolio optimization by separating asset selection from weight optimization, while leveraging three key structural properties of the Sharpe ratio (Kim & Lee, 2016).

**Proposition 3.1.** *For any $\lambda > 0$, the Sharpe ratio is scale-invariant with respect to $w$, i.e.,*

$$SR(w \mid \mu, \Sigma) = SR(\lambda w \mid \mu, \Sigma).$$

As a direct consequence, the budget constraint $\mathbf{1}^\top w = 1$ in Eq. (2) can be removed without changing the optimal Sharpe ratio value.

**Proposition 3.2.** *Let $w_1, w_2 \in \mathbb{R}^n$ be any two portfolios on $n$ assets, and let $\hat{w} \in \mathbb{R}^n$ be the tangent portfolio from Eq. (1) based on the market expected excess return $\mu$ and the market covariance $\Sigma$. Then,*

$$SR(w_1 \mid \mu, \Sigma) \geq SR(w_2 \mid \mu, \Sigma)$$

*if and only if*

$$\theta_1 \leq \theta_2$$

*where*

$$\theta_i := \arccos\left( \frac{(L_\Sigma^\top w_i)^\top (L_\Sigma^\top \hat{w})}{\|L_\Sigma^\top w_i\|_2 \|L_\Sigma^\top \hat{w}\|_2} \right)$$

*which denotes the angle between $L_\Sigma^\top w_i$ and $L_\Sigma^\top \hat{w}$ where $L_\Sigma$ is the Cholesky factor of $\Sigma$ (i.e., $\Sigma = L_\Sigma L_\Sigma^\top$).*

According to Proposition 3.2, we can alternatively compare portfolio Sharpe ratios via the angle $\theta$ to the tangency portfolio $\hat{w}$, where a smaller angle implies a larger Sharpe ratio. Let $\mathcal{K}$ denote the collection of all $k$-subsets of $\{1, \ldots, n\}$ and define:

$$P_K(\mathbb{R}^n) = \{w \in \mathbb{R}^n \mid w_i = 0, \; \forall i \notin K\}.$$

Then the original problem can be equivalently written as minimizing this angle as:

$$\min_{K \in \mathcal{K}} \left( \min_{w \in P_K\left(L_\Sigma^\top(\mathbb{R}^n)\right)} \arccos\left( \frac{(L_\Sigma^\top w)^\top (L_\Sigma^\top \hat{w})}{\|L_\Sigma^\top w\|_2 \|L_\Sigma^\top \hat{w}\|_2} \right) \right) \quad (3)$$

where

$$P_K(L_\Sigma^\top(\mathbb{R}^n)) = \{w \in \mathbb{R}^n \mid (L_\Sigma^\top w)_i = 0, \; \forall i \notin K\}.$$

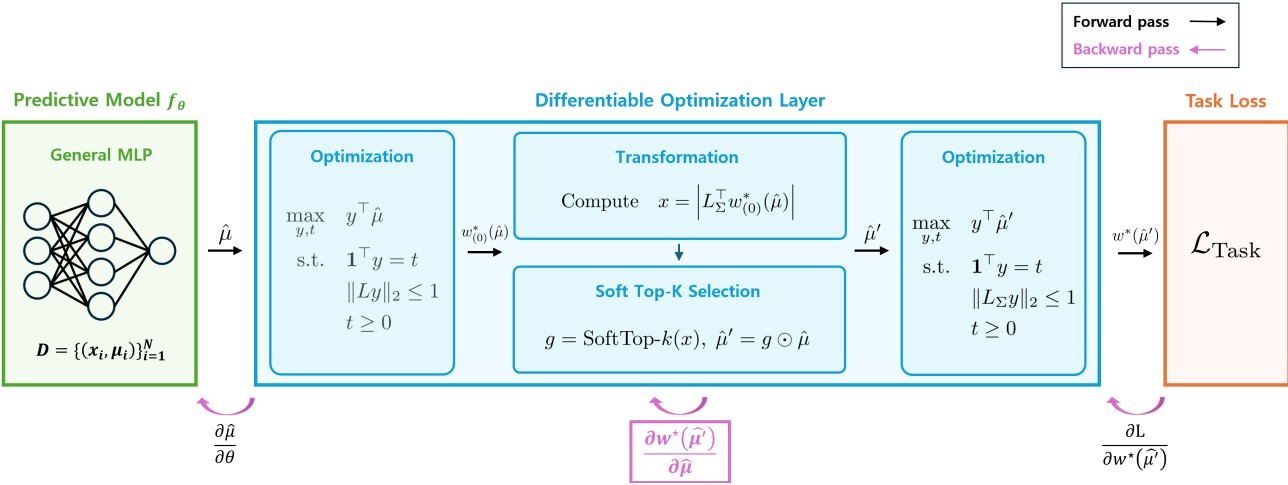

*Figure 1.* Overview of the proposed differentiable decision layer. Given predicted returns $\hat{\mu}$, the layer performs optimize $\to$ score transform $\to$ soft top-$k$ selection $\to$ re-optimize to produce sparse tangent portfolio weights, which define the task loss. Backpropagation flows through all steps, enabling end-to-end DFL. A central technical challenge is differentiating the optimization-based decision layer with respect to the prediction $\hat{\mu}$.

**Proposition 3.3.** *Let $K^* \in \mathcal{K}$ be a subset corresponding to the indices of the $k$ largest elements of $|L_\Sigma^\top \hat{w}|$, where $|\cdot|$ is the element-wise absolute value operator. Then $K^*$ is an optimal solution to Eq. (3), i.e.,*

$$K^* = \arg\min_{K \in \mathcal{K}} \left( \min_{w \in P_K\left(L_\Sigma^\top(\mathbb{R}^n)\right)} \arccos\left( \frac{(L_\Sigma^\top w)^\top (L_\Sigma^\top \hat{w})}{\|L_\Sigma^\top w\|_2 \|L_\Sigma^\top \hat{w}\|_2} \right) \right).$$

It follows from Proposition 3.3 that the optimizer of Eq. (3) selects the $k$ largest components of $|L_\Sigma^\top \hat{w}|$. Overall, OS-CAR consists of three stages. It first solves the tangency portfolio in Eq. (1) to obtain $\hat{w}$. Using this solution, it identifies a sparse support by selecting the $k$ assets corresponding to the $k$ largest entries of $|L_\Sigma^\top \hat{w}|$. Finally, it re-optimizes the Sharpe ratio objective restricted to the selected assets and derives the final optimal weight $w^*$. Proofs of Propositions 3.2 and 3.3 are provided in Appendix A for completeness.

### 3.2. Gradients for Decision-focused Learning

A common formulation views the underlying decision problem as a parametric constrained optimization problem, where a solution map $x^\star(c)$ returns the optimal decision for parameters $c$. A standard example is regret $\mathcal{R}$, which measures the suboptimality of the decision under perfect information as $\mathcal{R} = f\left(x^\star(c), c\right) - f\left(x^\star(\hat{c}), c\right)$, where $f(x, c)$ denotes the task objective to be maximized for decision $x$ evaluated under the true parameters $c$, and $\hat{c}$ is the prediction produced by the model. By differentiating this loss through the optimization layer and updating the prediction model accordingly, DFL learns to produce parameter estimates that are explicitly tuned for downstream performance. The

gradient can be calculated via the chain rule as

$$\frac{\partial \mathcal{R}}{\partial \theta} = \frac{\partial \mathcal{R}}{\partial x^\star(\hat{c})} \cdot \frac{\partial x^\star(\hat{c})}{\partial \hat{c}} \cdot \frac{\partial \hat{c}}{\partial \theta} \tag{4}$$

where $\theta$ denotes the parameters of the prediction model. The first and last terms in Eq. (4) are straightforward to differentiate, whereas the middle term $\partial x^\star(\hat{c})/\partial \hat{c}$ poses a substantial challenge, as the solution map is defined implicitly by an optimization problem and often lacks a closed-form expression. Moreover, for many practically relevant models, the mapping $\hat{c} \mapsto x^\star(\hat{c})$ can be non-smooth or even piecewise constant, leading to zero or undefined gradients on large regions of the parameter space.

A common approach to addressing this challenge is the use of differentiable convex optimization layers when the downstream problem is convex. Frameworks such as OptNet and CVXPYlayers provide principled gradients by differentiating through the solver (Agrawal et al., 2019). To enable such differentiation, the optimization problem must satisfy the rules of Disciplined Convex Programming (DCP), and when problem data are treated as learnable inputs, the formulation must also obey Disciplined Parametrized Programming (DPP), ensuring affine dependence on parameters and well-behaved solution maps.

## 4. Methodologies

### 4.1. Differentiable Reformulation of Sharpe Ratio Maximization

We now turn to the methodological core of our approach. Our end-to-end decision framework follows a select-and-reoptimize pipeline, but is designed to be fully differentiable

for learning. To embed Eq. (1) as a differentiable decision layer, the optimization problem must satisfy the rules of DPP. However, the standard Sharpe ratio formulation is non-convex and cannot be expressed using valid DPP atoms, preventing direct differentiation through a solver layer.

To obtain a tractable optimization program that is compatible with DPP, we follow the classical homogenization technique (Iyengar & Kang, 2005). Introducing a scaling variable $t \geq 0$ and defining $y = tw$, Eq. (1) can be equivalently rewritten as the convex Quadratically Constrained Quadratic Programming (QCQP) problem. Note that this QCQP is convex, but still not DPP-compliant. The difficulty arises because the problem data $\Sigma$ appear inside the nonlinear term $\sqrt{y^\top \Sigma y}$, violating the requirement that parameters must enter the program affinely. Since the covariance matrix $\Sigma$ is positive semidefinite, we apply its Cholesky factorization $\Sigma = L_\Sigma L_\Sigma^\top$. Substituting this factorization into the QCQP yields a formulation that satisfies the requirements of DPP, enabling reliable differentiation of the solution map through CVXPYlayers and allowing the optimization step to be embedded within DFL as shown in Eq. (5).

$$
\begin{array}{ll}
\max\limits_{y,t} & y^\top \mu \\
\text{s.t.} & \mathbf{1}^\top y = t \\
& \sqrt{y^\top \Sigma y} \leq 1 \\
& t \geq 0
\end{array}
\implies
\begin{array}{ll}
\max\limits_{y,t} & y^\top \mu \\
\text{s.t.} & \mathbf{1}^\top y = t \\
& \|L_\Sigma y\|_2 \leq 1 \\
& t \geq 0
\end{array}
\tag{5}
$$

## 4.2. Differentiable Top-$k$ Operator

In the asset selection step, we classify assets in the Cholesky-transformed space by computing $x = |L_\Sigma^\top w| \in \mathbb{R}^n$ and selecting the top-$k$ indices. A hard top-$k$ produces a $k$-hot mask but is discontinuous, and thus gradients do not flow through the selection step. To address this, we replace it with a simple differentiable top-$k$ operator that behaves like a soft $k$-hot mask and still enforces a strict sum–to–$k$ constraint (Ahle, 2022).

**Forward pass**  Let $x \in \mathbb{R}^n$ be the vector whose top-$k$ entries we want to select. In our model, $x = |L_\Sigma^\top w|$. Let $\sigma : \mathbb{R} \to (0,1)$ denote the logistic sigmoid, which is a smooth, monotone function. We then look for a scalar shift $t \in \mathbb{R}$ such that:

$$
\sum_{i=1}^n p_i(x) = k, \quad p_i(x) = \sigma(x_i + t) \quad i = 1, \ldots, n. \tag{6}
$$

Because $\sigma$ is monotone, the left-hand side of Eq. (6) is monotone in $t$ as well. The unique solution $t^*(x)$ can thus be obtained by a one-dimensional bisection search between two bounds at which the sigmoid is saturated. The result $p(x) \in (0,1)^n$ behaves like a softened top-$k$ mask: the $k$ largest coordinates of $x$ receive values close to 1, the others close to 0, and the total mass is exactly $k$.

**Backward pass**  The mapping $x \mapsto p(x)$ is differentiable, but $p_i(x) = \sigma(x_i + t(x))$ depends on $x$ both directly through $x_i$ and indirectly through the shared shift $t(x)$. For a fixed index $j$, the chain rule gives:

$$
\frac{\partial p_i(x)}{\partial x_j} = \sigma'(x_i + t(x))\left(\delta_{ij} + \frac{\partial t(x)}{\partial x_j}\right) \tag{7}
$$

where $\delta_{ij}$ is the Kronecker delta. Thus, the only missing piece is $\partial t(x)/\partial x_j$. We obtain $\partial t(x)/\partial x_j$ by differentiating the normalization constraint Eq. (6) with respect to $x_j$. Then we get:

$$
0 = \sum_{i=1}^n \sigma'(x_i + t(x))\left(\delta_{ij} + \frac{\partial t(x)}{\partial x_j}\right). \tag{8}
$$

Define:

$$
v_i(x) := \sigma'(x_i + t(x)) \quad \text{and} \quad S(x) := \sum_{i=1}^n v_i(x).
$$

Then Eq. (8) can be written as:

$$
\frac{\partial t(x)}{\partial x_j} = -\frac{v_j(x)}{S(x)}. \tag{9}
$$

Substituting Eq. (9) into Eq. (7) shows that the Jacobian has a diagonal minus rank-one structure:

$$
J(x) = \text{diag}(v(x)) - \frac{v(x)\,v(x)^\top}{S(x)}. \tag{10}
$$

Backpropagation does not require the full matrix $J(x)$. It only needs the vector–Jacobian product with an incoming gradient $g \in \mathbb{R}^n$:

$$
g^\top J(x) = g \odot v(x) - \frac{\langle g, v(x)\rangle}{S(x)}\, v(x) \tag{11}
$$

where $\odot$ denotes elementwise multiplication. This expression reuses the same shift $t(x)$ computed in the forward pass, and can be implemented without explicitly forming any large matrices, making the backward pass both numerically stable and efficient. In our implementation, we further introduce a sharpness parameter $\beta > 0$ to control the hardness of the soft top-$k$ operator. Instead of using $p_i(x) = \sigma(x_i + t)$, we define:

$$
p_i(x) = \sigma(\beta(x_i + t(x))), \qquad i = 1, \ldots, n \tag{12}
$$

where a larger value of $\beta$ produces a steeper transition and approximates the hard top-$k$ mask more closely.

This differentiable top-$k$ operator has several properties that make it well-suited for our framework. By construction, it satisfies the cardinality constraint, ensuring that the soft mask always preserves the desired sparsity budget. It is

also computationally efficient in both directions: the forward pass identifies the required shift via a lightweight one-dimensional bisection search, and the backward pass uses the closed-form vector-Jacobian product derived in Eq. (11), enabling fast and stable gradient evaluation. These properties enable the asset-selection module to be easily integrated into the end-to-end pipeline and allow the entire architecture to be trained with standard gradient-based optimizers.

We now describe how the proposed decision layer is embedded into a DFL framework for cardinality-constrained Sharpe ratio maximization. Figure 1 provides an overview of the proposed framework. A predictive model $f_\theta$ parametrized by $\theta$ takes as input a window of historical market data and outputs the quantities required by the optimization layer, such as a predicted mean return vector $\hat\mu$. These predictions are then passed to the differentiable decision layer, which produces a sparse portfolio $w^\star(\hat\mu)$ that satisfies the relevant budget, risk, and $k$-asset selection constraints.

The key role of the decision layer emerges during training: we define the learning objective directly in terms of downstream portfolio quality. By backpropagating through the differentiable decision mapping, the parameters $\theta$ are updated so that the predictive model learns to produce predictions aligned with the portfolio objective, rather than with a forecasting loss. As a result, the learned predictor is aligned with out-of-sample risk-adjusted performance under explicit cardinality constraints. The overall training procedure is summarized in Algorithm 1.

## 5. Experiments

### 5.1. Loss Function

We describe the training loss used when applying DFL to Sharpe ratio maximization. The decision-focused component is a regret-style loss based on the maximization problem in the objective of Eq. (5). Given the true mean returns $\mu^*$ and the predicted mean returns $\hat\mu$, we compare the portfolio $w^*(\hat\mu)$ that is optimal under the prediction with the portfolio $w^*(\mu^*)$ that is optimal under the ground-truth parameters. The resulting loss is:

$$\begin{aligned}
\mathcal{L}_{\text{DFL}} &= \mathcal{R}\big(w^*(\hat\mu), \mu^*\big) \\
&= f\big(w^*(\mu^*), \mu^*\big) - f\big(w^*(\hat\mu), \mu^*\big) \qquad (13) \\
&= \mu^{*\top} w^*(\mu^*) - \mu^{*\top} w^*(\hat\mu).
\end{aligned}$$

Since the first term on the right-hand side is constant with respect to the model parameters, we omit it from the loss to reduce computational cost.

Purely decision-focused loss can suffer from noisy or vanishing gradients, especially when the optimization landscape is relatively flat around the optimum. To alleviate this limita-

---

**Algorithm 1** Training predictive model $f_\theta$ with a Differentiable Decision Layer

---

**Input:** training set $\{(x_i, y_i)\}_{i=1}^N$, where $x_i$ is the look-back window of past returns and $y_i$ is the next-period return; learning rate $\eta$, batch size $B$, cardinality $k$, mixing weight $\alpha$.
Compute rolling covariance matrix $\Sigma_i$ from the look-back window $x_i$.
**for** each training epoch **do**
  **for** mini-batch $\mathcal{B} \subset \{1, \ldots, N\}$ with $|\mathcal{B}| = B$ **do**
    **for each** $(x_i, y_i) \in \mathcal{B}$ **do**
      $\hat\mu_i \leftarrow f_\theta(x_i)$
      $w_i^{(0)} \leftarrow w^*(\hat\mu_i; \Sigma_i)$ **by solving** Eq. (5)
      $s_i \leftarrow \big|L_{\Sigma_i}^\top w_i^{(0)}\big|$
      $\mu_i' \leftarrow \text{SoftTop-}k(\hat\mu_i; s_i, k)$
      $w_i^* \leftarrow w^*(\mu_i'; \Sigma_i)$ **by solving** Eq. (5)
      $\mathcal{L}_{\text{DFL}}^{(i)} \leftarrow -y_i^\top w_i^*$
      $\mathcal{L}_{\text{MSE}}^{(i)} \leftarrow \|\hat\mu_i - y_i\|_2^2$
      $\mathcal{L}^{(i)} \leftarrow \alpha \mathcal{L}_{\text{DFL}}^{(i)} + (1-\alpha)\mathcal{L}_{\text{MSE}}^{(i)}$
    **end for**
    $\mathcal{L} \leftarrow \frac{1}{|\mathcal{B}|}\sum_{i\in\mathcal{B}} \mathcal{L}^{(i)}$
    $\theta \leftarrow \theta - \eta\nabla_\theta \mathcal{L}$
  **end for**
**end for**

---

tion, we augment the regret loss with a standard prediction term that penalizes the squared error between $\hat\mu$ and $\mu^*$. Concretely, we train the network using the task loss $\mathcal{L}_{\text{Task}}$:

$$\mathcal{L}_{\text{Task}} = \alpha\mathcal{L}_{\text{DFL}} + (1-\alpha)\mathcal{L}_{\text{MSE}} \qquad (14)$$

where $\alpha \in [0, 1]$ controls the balance between decision quality and prediction accuracy. Note that the MSE loss is defined as $\mathcal{L}_{\text{MSE}} = \frac{1}{N}\sum_{i=1}^N \|\mu_i^* - \hat\mu_i\|_2^2$. This combined loss leads to more stable training and encourages the model to produce forecasts that are both accurate and well aligned with the downstream portfolio optimization task.

### 5.2. Experimental Setup

We use a 10-year daily closing price dataset from Yahoo Finance covering four markets: EuroStoxx50, FTSE100, KOSPI200, and Nikkei225. The sample period spans from January 2016 to December 2025. We restrict attention to constituents with stable index membership, omitting any firms that enter or leave the index during the period.

For the prediction model, we first preprocess the time series into rolling windows of daily returns with a 100-day look-back horizon. Each input feature vector $X_t$ consists of the past 100 days of returns up to day $t$, and the network is trained to predict the one-step-ahead return $\hat\mu_{t+1}$. This predicted mean vector is then passed to the optimization

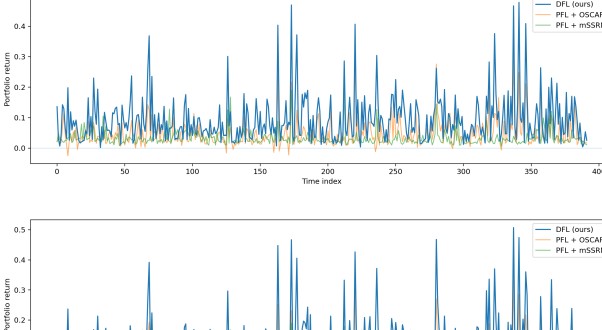

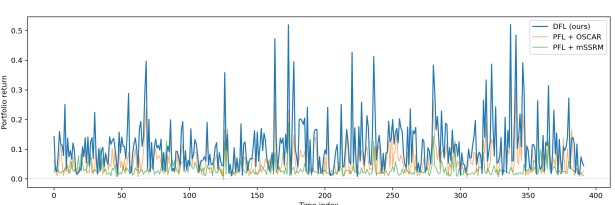

*Figure 2.* Portfolio return time-series of FTSE100 on the test set across three cardinality levels $\rho \in \{10\%, 15\%, 20\%\}$. We compare our DFL framework with the PFL baselines. Across most time periods, DFL attains higher returns than the competing methods.

layer, which solves the portfolio problem and outputs the corresponding optimal weights $w^*(\hat{\mu}_{t+1})$. The full dataset is split chronologically into training and test sets with an 80/20 ratio. Throughout all experiments, the input covariance matrix is estimated in a rolling-window manner for each sample, using the same historical look-back window that forms the model input. Once computed for a given sample, covariance is held fixed within the corresponding downstream optimization layer and is not learned or predicted by the neural network. In other words, the covariance structure varies across samples through the rolling window, but remains exogenous with respect to the predictive model parameters. This design avoids feeding noisy model outputs into the risk constraints of the downstream optimization, which can otherwise lead to unstable or infeasible problem instances, while still allowing the risk estimate to reflect local market conditions.

## 5.3. Baselines and Evaluation Metric

We evaluate our proposed framework against two conventional learning-optimization frameworks and three state-of-the-art portfolio optimization models for cardinality-constrained Sharpe ratio maximization.

**Conventional frameworks**

1. **Historic**: In the historic baseline, we do not train a prediction model and instead use the historical expected return vector $\mu$ directly in the optimization problem, i.e., the portfolio weights are obtained as $w^*(\mu)$ from the mean-variance objective.

2. **PFL**: In the PFL framework, we first learn a predictive model $f_\theta$ and use its output $\hat{\mu} = f_\theta(x)$ as the expected return in the downstream optimization, yielding portfolio weights $w^*(\hat{\mu})$.

**Optimization models** For the optimization, we consider three state-of-the-art models for cardinality-constrained Sharpe ratio maximization:

1. **OSCAR** (Bae et al., 2025): A three-stage heuristic that solves the tangency portfolio, selects the top-$k$ assets in the Cholesky-transformed space, and re-optimizes weights over the selected support.

2. **SD-relaxation** (Kim et al., 2016): This approach relaxes the cardinality-constrained Sharpe ratio maximization into a tractable semidefinite program by lifting the portfolio weights to a positive semidefinite matrix and replacing the nonconvex sparsity structure with a convex surrogate.

3. **mSSRM-PGA** (Lin et al., 2024): This method tackles $m$-sparse Sharpe ratio maximization by first rewriting the fractional objective into an equivalent $m$-sparse quadratic program, and then running a proximal-gradient algorithm whose proximal step simply keeps the largest positive entries (up to the sparsity budget) and truncates the rest to zero.

**Evaluation metric** Our primary evaluation metric is the out-of-sample Sharpe ratio computed from the realized portfolio returns. For a given sequence of portfolio weights $\{w_t\}_{t=1}^T$ and next-period realized returns $\{r_{t+1}\}_{t=1}^T$, we form the realized return series:

$$r_{p,t} = w_t^\top r_{t+1}, \qquad t = 1, \ldots, T.$$

The (daily) Sharpe ratio is then defined as:

$$\mathrm{SR} = \frac{\bar{r}_p}{s_p} = \frac{\frac{1}{T}\sum_{t=1}^T r_{p,t}}{\sqrt{\frac{1}{T-1}\sum_{t=1}^T (r_{p,t} - \bar{r}_p)^2}}$$

where $\bar{r}_p$ and $s_p$ denote the sample mean and sample standard deviation of the realized portfolio returns. We report this Sharpe ratio for every method and market as the main measure of risk-adjusted performance. Higher Sharpe ratios indicate better return–volatility trade-offs.

*Table 1.* Out-of-sample Sharpe ratios on four equity markets. Values are reported as mean and standard deviation over five random seeds. The Historic results are estimated using the full training set and thus have zero standard deviation. The best-performing method for each market-cardinality pair is bold-lettered.

| Market | $N$ | $k$ | Historic | | | PFL | | | DFL |
| --- | --- | --- | --- | --- | --- | --- | --- | --- | --- |
| | | | OSCAR | SD-relaxation | mSSRM | OSCAR | SD-relaxation | mSSRM | Ours |
| EuroStoxx50 | 47 | 5 | $0.156 \pm 0.000$ | $0.077 \pm 0.000$ | $0.148 \pm 0.000$ | $0.809 \pm 0.025$ | $\mathbf{0.982 \pm 0.087}$ | $0.821 \pm 0.074$ | $0.955 \pm 0.014$ |
| | | 7 | $0.198 \pm 0.000$ | $-0.037 \pm 0.000$ | $0.149 \pm 0.000$ | $0.827 \pm 0.023$ | $\mathbf{1.041 \pm 0.066}$ | $0.846 \pm 0.089$ | $0.972 \pm 0.016$ |
| | | 9 | $0.228 \pm 0.000$ | $-0.045 \pm 0.000$ | $0.152 \pm 0.000$ | $0.841 \pm 0.020$ | $\mathbf{1.116 \pm 0.058}$ | $0.889 \pm 0.081$ | $0.983 \pm 0.016$ |
| FTSE100 | 93 | 9 | $0.269 \pm 0.000$ | $0.198 \pm 0.000$ | $0.167 \pm 0.000$ | $0.776 \pm 0.025$ | $0.931 \pm 0.052$ | $0.782 \pm 0.044$ | $\mathbf{0.983 \pm 0.016}$ |
| | | 14 | $0.322 \pm 0.000$ | $0.245 \pm 0.000$ | $0.234 \pm 0.000$ | $0.853 \pm 0.022$ | $\mathbf{1.073 \pm 0.061}$ | $0.841 \pm 0.049$ | $1.049 \pm 0.034$ |
| | | 19 | $0.396 \pm 0.000$ | $0.254 \pm 0.000$ | $0.237 \pm 0.000$ | $0.888 \pm 0.020$ | $\mathbf{1.142 \pm 0.057}$ | $0.873 \pm 0.047$ | $1.071 \pm 0.021$ |
| KOSPI200 | 162 | 16 | $0.426 \pm 0.000$ | $0.727 \pm 0.000$ | $0.319 \pm 0.000$ | $0.887 \pm 0.017$ | $1.182 \pm 0.042$ | $1.372 \pm 0.052$ | $\mathbf{1.958 \pm 0.022}$ |
| | | 24 | $0.485 \pm 0.000$ | $0.814 \pm 0.000$ | $0.332 \pm 0.000$ | $0.918 \pm 0.010$ | $1.684 \pm 0.037$ | $1.491 \pm 0.058$ | $\mathbf{2.030 \pm 0.062}$ |
| | | 32 | $0.527 \pm 0.000$ | $0.845 \pm 0.000$ | $0.325 \pm 0.000$ | $0.935 \pm 0.009$ | $1.793 \pm 0.118$ | $1.586 \pm 0.171$ | $\mathbf{2.098 \pm 0.016}$ |
| Nikkei225 | 208 | 21 | $0.258 \pm 0.000$ | $0.198 \pm 0.000$ | $0.273 \pm 0.000$ | $0.567 \pm 0.014$ | $0.743 \pm 0.103$ | $0.614 \pm 0.092$ | $\mathbf{0.862 \pm 0.060}$ |
| | | 31 | $0.284 \pm 0.000$ | $0.243 \pm 0.000$ | $0.276 \pm 0.000$ | $0.620 \pm 0.012$ | $0.812 \pm 0.111$ | $0.662 \pm 0.094$ | $\mathbf{0.878 \pm 0.040}$ |
| | | 42 | $0.295 \pm 0.000$ | $0.248 \pm 0.000$ | $0.280 \pm 0.000$ | $0.649 \pm 0.011$ | $0.886 \pm 0.126$ | $0.721 \pm 0.107$ | $\mathbf{0.946 \pm 0.130}$ |

## 6. Results

Table 1 summarizes the out-of-sample Sharpe ratios obtained across the four equity markets and cardinality levels $\rho \in \{10\%, 15\%, 20\%\}$. Note that our objective is to maximize the Sharpe ratio. The cardinality $k$ is derived by multiplying each level by the number of assets $N$ and rounding to the nearest integer as $k = \mathrm{round}(\rho \cdot N)$. The Historic baselines, which directly optimize on sample mean returns without any learning component, consistently underperform. This confirms that purely backward-looking estimates are insufficient in non-stationary markets.

PFL delivers substantial improvements over Historic, but its performance varies substantially depending on the optimization model. In EuroStoxx50 and partly in FTSE100, the SD-relaxation-based PFL baseline achieves slightly higher average Sharpe ratios than the proposed framework. However, our method remains highly competitive in these cases, with only small performance gaps, while exhibiting smaller standard deviations across seeds. This suggests that although SD-relaxation can occasionally obtain strong average performance, its results are relatively more sensitive to training randomness.

In contrast, the advantage of the proposed framework becomes much clearer in larger asset universes. On KOSPI200 and Nikkei225, the gap between our method and the PFL baselines widens noticeably across most cardinality levels, particularly for KOSPI200 where our framework consistently achieves substantially higher Sharpe ratios than all PFL variants. These results demonstrate that aligning prediction, selection, and re-optimization through end-to-end training is highly effective for Sharpe ratio maximization under sparsity constraints.

In addition to the Sharpe ratio, we further compare the portfolio returns achieved by each model over the test period.

Figure 2 presents illustrative portfolio return time-series results on FTSE100 across different cardinality levels. The results show that our DFL framework consistently attains higher portfolio returns than the PFL baselines across most time periods. Similar patterns are also observed in the other equity markets, as reported in Appendix C.3. At the same time, because our current formulation allows short selling, higher Sharpe ratios may be accompanied by a more aggressive downside-risk profile. To examine this aspect, Appendix C.2 reports additional Maximum Drawdown (MDD) results and discusses the risk trade-off of the learned portfolios.

Overall, the evidence shows that decision-focused training can optimize predictions directly for the sparse Sharpe ratio objective through a differentiable decision layer. This produces sparse portfolios that outperform Historic and PFL baselines, particularly in larger asset universes, while remaining competitive in smaller markets.

## 7. Conclusion

We present a DFL framework for Sharpe ratio maximization under cardinality constraints. Our approach enables end-to-end training by differentiating through the sparse decision pipeline, allowing gradients to propagate through prediction, soft top-$k$ selection, and re-optimization. As a result, the learning objective is directly aligned with downstream portfolio quality, in contrast to traditional PFL approaches that rely on forecasting loss as a proxy.

Extensive experiments confirm that the proposed framework provides an effective way to train predictive models for sparse Sharpe ratio portfolio construction. Rather than treating return prediction as a separate intermediate task, our approach aligns the predictor with the downstream selection and re-optimization procedure. The empirical results show

that this DFL alignment leads to competitive and often superior risk-adjusted performance, especially in larger asset universes where the sparse selection problem becomes more challenging.

Despite these advantages, the framework has three main limitations. First, differentiating through the sparse selection module, especially the soft top-$k$ operator, introduces computational overhead as the asset universe or the sparsity budget grows. Second, our framework allows short selling and does not impose practical constraints such as long-only positions, leverage limits, or turnover penalties. Lastly, our approach is tailored to sparse tangent portfolio optimization, and extending it to other cardinality-constrained portfolio objectives with different downstream goals is nontrivial.

Future work includes designing more efficient differentiable selection mechanisms to improve scalability, incorporating practical constraints to make the framework more realistic, and exploring broader applications where DFL can improve decision quality beyond sparse portfolio construction.

## Impact Statement

This paper presents work whose goal is to advance the field of Machine Learning in Finance. There are many potential societal consequences of our work, none of which we feel must be specifically highlighted here.

## Acknowledgement

This work was supported by the National Research Foundation (NRF) of Korea grant funded by the Ministry of Science and ICT (MSIT) of Korea (No. RS-2022-NR068758, No. RS-2025-02216640, No. RS-2025-24803208) and the Institute of Information & Communications Technology Planning & Evaluation (IITP) grant funded by the Ministry of Science and ICT (MSIT) of Korea (No. RS-2020-II201336, Artificial Intelligence Graduate School Program (UNIST)).

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

# A. Additional Proofs

The following proofs are included to make the paper self-contained. They follow the geometric Sharpe ratio arguments of Kim & Lee (2016) and the OSCAR derivation of Bae et al. (2025), with notation adapted to the present paper.

## A.1. Proof of Proposition 3.2

Let $w^* = \hat{w}$ denote the tangent portfolio. First note that the tangent portfolio is given by

$$w^* = \arg\max_{w \in \mathbb{R}^n} \mathrm{SR}(w \mid \mu, \Sigma) = \Sigma^{-1}\mu,$$

up to a positive scaling factor, and the maximum Sharpe ratio is

$$
\begin{aligned}
\mathrm{SR}(w^* \mid \mu, \Sigma) &= \frac{\mu^\top w^*}{\sqrt{w^{*\top}\Sigma w^*}} \\
&= \frac{\mu^\top (\Sigma^{-1}\mu)}{\sqrt{\mu^\top \Sigma^{-1}\Sigma\Sigma^{-1}\mu}} \\
&= \frac{\mu^\top \Sigma^{-1}\mu}{\sqrt{\mu^\top \Sigma^{-1}\mu}} \\
&= \sqrt{\mu^\top \Sigma^{-1}\mu}.
\end{aligned}
$$

Then,

$$
\begin{aligned}
\theta_1 &= \arccos\left(\frac{(L_\Sigma^\top w_1)^\top (L_\Sigma^\top w^*)}{\|L_\Sigma^\top w_1\|_2 \|L_\Sigma^\top w^*\|_2}\right) \\
&= \arccos\left(\frac{w_1^\top L_\Sigma L_\Sigma^\top w^*}{\sqrt{(L_\Sigma^\top w_1)^\top (L_\Sigma^\top w_1)}\sqrt{(L_\Sigma^\top w^*)^\top (L_\Sigma^\top w^*)}}\right) \\
&= \arccos\left(\frac{w_1^\top \Sigma w^*}{\sqrt{w_1^\top \Sigma w_1}\sqrt{w^{*\top}\Sigma w^*}}\right) \\
&= \arccos\left(\frac{w_1^\top \mu}{\sqrt{w_1^\top \Sigma w_1}\sqrt{\mu^\top \Sigma^{-1}\mu}}\right) \\
&= \arccos\left(\frac{\mathrm{SR}(w_1 \mid \mu, \Sigma)}{\mathrm{SR}(w^* \mid \mu, \Sigma)}\right),
\end{aligned}
$$

and similarly,

$$\theta_2 = \arccos\left(\frac{\mathrm{SR}(w_2 \mid \mu, \Sigma)}{\mathrm{SR}(w^* \mid \mu, \Sigma)}\right).$$

Since $\arccos(x)$ is a decreasing function of $x$, we have

$$\theta_1 \leq \theta_2 \quad \text{if and only if} \quad \mathrm{SR}(w_1 \mid \mu, \Sigma) \geq \mathrm{SR}(w_2 \mid \mu, \Sigma).$$

This proves the proposition. □

## A.2. Proof of Proposition 3.3

Let

$$L_\Sigma^\top \hat{w} = (x_1, x_2, \ldots, x_n) \in \mathbb{R}^n.$$

Without loss of generality, assume

$$|x_1| \geq |x_2| \geq \cdots \geq |x_n|.$$

Then, $K^* = \{1, 2, \ldots, k\}$.

Since $\arccos$ is a monotonic decreasing function on $[0, 1]$,

$$\arg\min_{K \in \mathcal{K}} \left( \min_{w \in P_K(L_\Sigma^\top(\mathbb{R}^n))} \arccos \left( \frac{(L_\Sigma^\top w)^\top (L_\Sigma^\top \hat{w})}{\|L_\Sigma^\top w\|_2 \|L_\Sigma^\top \hat{w}\|_2} \right) \right)$$

$$= \arg\max_{K \in \mathcal{K}} \left( \max_{w \in P_K(L_\Sigma^\top(\mathbb{R}^n))} \frac{(L_\Sigma^\top w)^\top (L_\Sigma^\top \hat{w})}{\|L_\Sigma^\top w\|_2 \|L_\Sigma^\top \hat{w}\|_2} \right).$$

Let $(L_\Sigma^\top \hat{w})_K$ be a projection where the component of $L_\Sigma^\top \hat{w}$ corresponding to the index not included in $K$ is changed to $0$. Then, we have

$$\arg\max_{K \in \mathcal{K}} \left( \max_{w \in P_K(L_\Sigma^\top(\mathbb{R}^n))} \frac{(L_\Sigma^\top w)^\top (L_\Sigma^\top \hat{w})}{\|L_\Sigma^\top w\|_2 \|L_\Sigma^\top \hat{w}\|_2} \right)$$

$$= \arg\max_{K \in \mathcal{K}} \left( \max_{w \in P_K(L_\Sigma^\top(\mathbb{R}^n))} \frac{(L_\Sigma^\top w)^\top (L_\Sigma^\top \hat{w})_K}{\|L_\Sigma^\top w\|_2 \|L_\Sigma^\top \hat{w}\|_2} \right)$$

$$= \arg\max_{K \in \mathcal{K}} \left( \max_{w \in P_K(L_\Sigma^\top(\mathbb{R}^n))} \frac{\|(L_\Sigma^\top \hat{w})_K\|_2}{\|L_\Sigma^\top \hat{w}\|_2} \frac{(L_\Sigma^\top w)^\top (L_\Sigma^\top \hat{w})_K}{\|L_\Sigma^\top w\|_2 \|(L_\Sigma^\top \hat{w})_K\|_2} \right)$$

$$= \arg\max_{K \in \mathcal{K}} \left( \frac{\|(L_\Sigma^\top \hat{w})_K\|_2}{\|L_\Sigma^\top \hat{w}\|_2} \frac{(L_\Sigma^\top \hat{w})_K^\top (L_\Sigma^\top \hat{w})_K}{\|(L_\Sigma^\top \hat{w})_K\|_2 \|(L_\Sigma^\top \hat{w})_K\|_2} \right)$$

$$(\because \ w = (L_\Sigma^\top)^{-1} (L_\Sigma^\top \hat{w})_K \in P_K(L_\Sigma^\top(\mathbb{R}^n)))$$

$$= \arg\max_{K \in \mathcal{K}} \left( \frac{\|(L_\Sigma^\top \hat{w})_K\|_2}{\|L_\Sigma^\top \hat{w}\|_2} \right)$$

$$= K^*.$$

Thus, $K^* = \{1, \ldots, k\}$ is the optimal solution. □

## B. Experiment Settings Details

### B.1. Experimental Setup

The experiments were implemented in Python, using PyTorch for the predictive model and CVXPYlayers for the optimization module (Agrawal et al., 2019), which enabled differentiation through the portfolio optimization step and thus an end-to-end decision-focused learning pipeline. The prediction network consists of two fully connected hidden layers with 512 and 256 neurons, respectively, trained with a learning rate of 0.001 and a mini-batch size of 64. The combined loss uses a weighting parameter $\alpha = 0.5$ to balance prediction accuracy and decision quality, which we found to improve training stability and efficiency. In the differentiable top-$k$ selection layer, the bisection procedure is run for 32 iterations.

Since the covariance matrix is estimated from rolling windows, it can occasionally become rank-deficient or numerically ill-conditioned. To ensure positive semidefiniteness and stable Cholesky factorization, we apply diagonal shrinkage to each rolling covariance matrix by shrinking it toward a scaled identity matrix, followed by a small eigenvalue-based jitter when necessary. Specifically, for each rolling covariance estimate $\hat{\Sigma}_t$, we use

$$\tilde{\Sigma}_t = (1 - \lambda)\hat{\Sigma}_t + \lambda \frac{\text{tr}(\hat{\Sigma}_t)}{n} I,$$

with $\lambda = 0.10$, and add a small diagonal jitter if needed to guarantee a valid Cholesky factorization. In addition, because short selling is allowed in the portfolio optimization layer, the learned portfolio weights can exhibit extreme fluctuations during training. To mitigate this numerical instability, we add a small offset to the QCQP scaling variable $c$ when recovering the portfolio weights. All experiments were carried out on a machine equipped with an Intel Xeon Silver 4310 CPU (@2.10GHz) and an NVIDIA RTX A6000 GPU with 48GB of memory, and repeated over 5 random seeds with mean and standard deviation reported, providing sufficient computational resources to train the model and solve the embedded optimization problems.

**B.2. Details of the SD-relaxation baseline**

Kim et al. (2016) relax the $k$-sparse Sharpe ratio maximization by lifting the weight vector to a positive semidefinite matrix variable $Y \succeq 0$ (a surrogate of $ww^\top$). With $M := \mu\mu^\top$, the squared numerator becomes a linear trace objective $\mathrm{Tr}(MY)$, while the risk normalization can be enforced by $\mathrm{Tr}(\Sigma Y) = 1$. The auxiliary scalar $z = \mathrm{Tr}(Y)$ captures the scale of the lifted variable, and sparsity is imposed via the convex surrogate $1^\top |Y| 1 \leq kz$ (originating from $\|w\|_1^2 \leq k\|w\|_2^2$ for $k$-sparse vectors). This yields the following tractable SDP:

$$\max_{Y,z} \quad \mathrm{Tr}(MY)$$

$$\begin{aligned}
\text{s.t.} \quad & \mathrm{Tr}(Y) = z \\
& \mathbf{1}^\top |Y| \, \mathbf{1} \leq kz \\
& \mathrm{Tr}(\Sigma Y) = 1 \\
& Y \succeq 0.
\end{aligned}$$

Since this optimization does not strictly enforce the target cardinality, a single run can yield a portfolio whose realized number of selected assets differs from our proposed framework. To ensure a fair comparison, we repeat the SD-relaxation baseline experiment while adjusting the sparsity parameter used by the baseline until the obtained portfolio attains the same realized cardinality as our proposed method. This procedure may correspond to using a looser effective cardinality level than the nominal $k$ in the baseline formulation, but it aligns the final number of selected assets across methods.

**B.3. Details of the mSSRM-PGA baseline**

Lin et al. (2024) solve the $m$-sparse Sharpe ratio maximization via a projected gradient (PGA) scheme that performs gradient ascent on the Sharpe ratio objective while enforcing sparsity at every step:

$$\max_w \quad \frac{\hat{\mu}^\top w}{\sqrt{w^\top \Sigma w}} \quad \text{s.t.} \quad \|w\|_0 \leq k. \tag{15}$$

Starting from an initialization $w^{(0)}$, the iterate is updated by a gradient step followed by a hard-thresholding projection onto the top-$k$ support:

$$w^{(t+1)} = \Pi_k \Big( w^{(t)} + \eta \, \nabla \mathrm{SR}\big(w^{(t)}\big) \Big) \tag{16}$$

where $\Pi_k(\cdot)$ keeps the $k$ largest entries in magnitude and sets the rest to zero.

## C. Additional Experiments and Results

### C.1. $\alpha$ Sensitivity

In the main experiments, we use the mixed training objective

$$\mathcal{L}_{\text{Task}} = \alpha \mathcal{L}_{\text{DFL}} + (1-\alpha)\mathcal{L}_{\text{MSE}}, \tag{17}$$

with $\alpha = 0.5$ as a default setting. Here, the MSE term is not introduced as a new loss function or as a separate methodological contribution. Rather, it is the standard prediction loss used as an auxiliary supervised signal to stabilize the decision-focused training procedure. Thus, the role of the mixed objective is not to claim a novel loss design, but to provide a stable default training formulation for the proposed differentiable sparse portfolio optimization pipeline.

This choice is motivated by the fact that pure decision-focused learning can suffer from weak, noisy, or locally unstable gradients. In our setting, this issue can be more pronounced because the downstream pipeline contains both a differentiable sparse-selection step and a convex re-optimization layer. As a result, the decision loss may provide gradients that are small in magnitude or poorly informative in some regions of the training landscape. By contrast, the MSE term provides a direct predictive supervision signal on the expected returns. Recent studies on prediction-loss-guided decision-focused learning provide a related geometric motivation. In particular, Jeon et al. (2025) analyze the Hessian eigenvalue density of prediction, decision, and mixed losses, and show that the eigenvalues of the decision loss are highly concentrated near zero, indicating a largely flat decision-loss landscape. In contrast, the prediction loss exhibits a broader eigenvalue spectrum, including

*Table 2.* $\alpha$ sensitivity case study on EuroStoxx50. Values are out-of-sample Sharpe ratios, reported as mean $\pm$ standard deviation.

| $\alpha$ | $\rho = 10\%$ | $\rho = 15\%$ | $\rho = 20\%$ |
|---|---|---|---|
| 0.0 | $0.809 \pm 0.025$ | $0.827 \pm 0.023$ | $0.841 \pm 0.020$ |
| 0.1 | $0.826 \pm 0.025$ | $0.852 \pm 0.024$ | $0.862 \pm 0.013$ |
| 0.5 | $0.955 \pm 0.014$ | $0.972 \pm 0.016$ | $0.983 \pm 0.016$ |
| 0.9 | $1.041 \pm 0.081$ | $1.084 \pm 0.083$ | $1.140 \pm 0.154$ |
| 1.0 | $1.165 \pm 0.106$ | $1.056 \pm 0.125$ | $0.940 \pm 0.097$ |

negative eigenvalues and several larger positive eigenvalues, while the convex combination of prediction and decision losses yields a more balanced spectrum. This suggests that the prediction loss can enrich the curvature and gradient signal of decision-focused objectives, especially when the pure decision loss provides weak or unstable gradients. Therefore, the mixed objective is used as a practical compromise that preserves decision-level alignment while improving optimization stability.

To examine the effect of $\alpha$, we conduct a small case study on EuroStoxx50. We vary $\alpha \in \{0, 0.1, 0.5, 0.9, 1.0\}$ and evaluate three cardinality levels, $\rho \in \{10\%, 15\%, 20\%\}$. Each setting is trained with five random seeds, using the same data split, model architecture, covariance estimation procedure, and optimization pipeline as in the main experiments. The case $\alpha = 0$ corresponds to pure prediction-focused training with MSE loss, whereas $\alpha = 1$ corresponds to pure decision-focused training.

Table 2 shows that the choice of $\alpha$ has a noticeable effect on performance. Very small values of $\alpha$, such as $\alpha = 0$ and $\alpha = 0.1$, are dominated by the MSE term and consistently produce lower Sharpe ratios across all cardinality levels. This suggests that prediction accuracy alone is insufficient to fully capture the downstream sparse portfolio objective. As $\alpha$ increases, the average Sharpe ratio generally improves, indicating the benefit of incorporating the decision-focused objective. However, larger values of $\alpha$ also lead to noticeably higher variability across seeds. For example, $\alpha = 0.9$ achieves the highest average Sharpe ratios for $\rho = 15\%$ and $\rho = 20\%$, but its standard deviations are substantially larger than those of $\alpha = 0.5$. Similarly, the pure DFL setting $\alpha = 1$ achieves the best average performance at $\rho = 10\%$, but its performance becomes less consistent as the cardinality level increases, and it underperforms $\alpha = 0.9$ at $\rho = 15\%$ and both mixed settings at $\rho = 20\%$.

Overall, these results suggest that larger decision-focused weights can improve average decision quality, but may also make training more sensitive to random initialization and mini-batch sampling. In contrast, $\alpha = 0.5$ provides a stable performance-variance trade-off: it consistently improves over the MSE-dominated settings while maintaining the smallest standard deviations across all cardinality levels. Therefore, we use $\alpha = 0.5$ as a stable default rather than a validation-tuned optimum. More broadly, the gains should be interpreted as coming from the full mixed training formulation, rather than from the decision-focused loss term in isolation.

### C.2. Additional Risk Metrics

The main experiments focus on the out-of-sample Sharpe ratio because our downstream objective is sparse Sharpe ratio maximization. However, the Sharpe ratio alone does not fully characterize the practical risk profile of the learned portfolios. In particular, it does not directly measure downside risk or the severity of cumulative losses over the investment horizon. To provide an additional diagnostic, we report Maximum Drawdown (MDD) for the learned portfolios.

Since our current formulation allows short-selling, the learned portfolios may take both long and short positions. Therefore, we compare DFL and PFL under the same OSCAR-based portfolio construction pipeline and examine whether decision-focused training leads to portfolios with a larger downside-risk profile. For each market and cardinality level, we report the MDD evaluated on the test period.

Table 3 shows that although DFL improves the downstream Sharpe ratio objective in the main experiments, it often leads to larger MDD values than PFL. This suggests that the DFL portfolios can be more aggressive under the current short-selling-allowed formulation. Therefore, the higher Sharpe ratios of DFL should be interpreted together with its increased drawdown risk. Overall, this analysis highlights an important limitation of the current formulation. Our contribution is primarily methodological: we show that decision-focused learning can be successfully applied to cardinality-constrained sparse Sharpe ratio optimization. Incorporating more practical constraints, such as long-only positions, leverage limits,

*Table 3.* Additional MDD metrics across four equity markets. MDD is evaluated on the test period. Values are reported as mean $\pm$ standard deviation over five random seeds.

| Market | Method | $\rho = 10\%$ | $\rho = 15\%$ | $\rho = 20\%$ |
|---|---|---|---|---|
| EuroStoxx50 | PFL | $0.474 \pm 0.257$ | $0.340 \pm 0.079$ | $0.317 \pm 0.113$ |
| | DFL | $0.310 \pm 0.105$ | $0.403 \pm 0.109$ | $0.390 \pm 0.055$ |
| FTSE100 | PFL | $0.497 \pm 0.095$ | $0.495 \pm 0.099$ | $0.497 \pm 0.102$ |
| | DFL | $0.710 \pm 0.100$ | $0.605 \pm 0.209$ | $0.607 \pm 0.129$ |
| KOSPI200 | PFL | $0.775 \pm 0.109$ | $0.790 \pm 0.121$ | $0.796 \pm 0.128$ |
| | DFL | $0.952 \pm 0.021$ | $0.945 \pm 0.033$ | $0.925 \pm 0.055$ |
| Nikkei225 | PFL | $0.484 \pm 0.192$ | $0.528 \pm 0.199$ | $0.558 \pm 0.202$ |
| | DFL | $0.543 \pm 0.095$ | $0.559 \pm 0.100$ | $0.599 \pm 0.136$ |

turnover penalties, or margin constraints, remains an important direction for future work.

### C.3. Additional Time-series Results

We provide three additional equity markets (EuroStoxx50, KOSPI200, and Nikkei225) comparisons across three cardinality levels 10%, 15%, and 20% in Figure 3, 4, and 5, reporting the portfolio return at each time step for our DFL framework and PFL baselines: OSCAR and mSSRM-PGA. The SD-relaxation baseline shows markedly higher volatility and occasional extreme swings depending on training, which is also reflected in the larger variability summarized in Table 1. Due to this excessive instability, we exclude SD-relaxation from the remaining three markets reported in the subsequent appendix results. Notably, the performance gap tends to widen as $k$ increases, suggesting that our decision-focused training becomes particularly effective when the allocation space is less restrictive, and the model can better exploit improved mean estimates for downstream risk-adjusted performance.

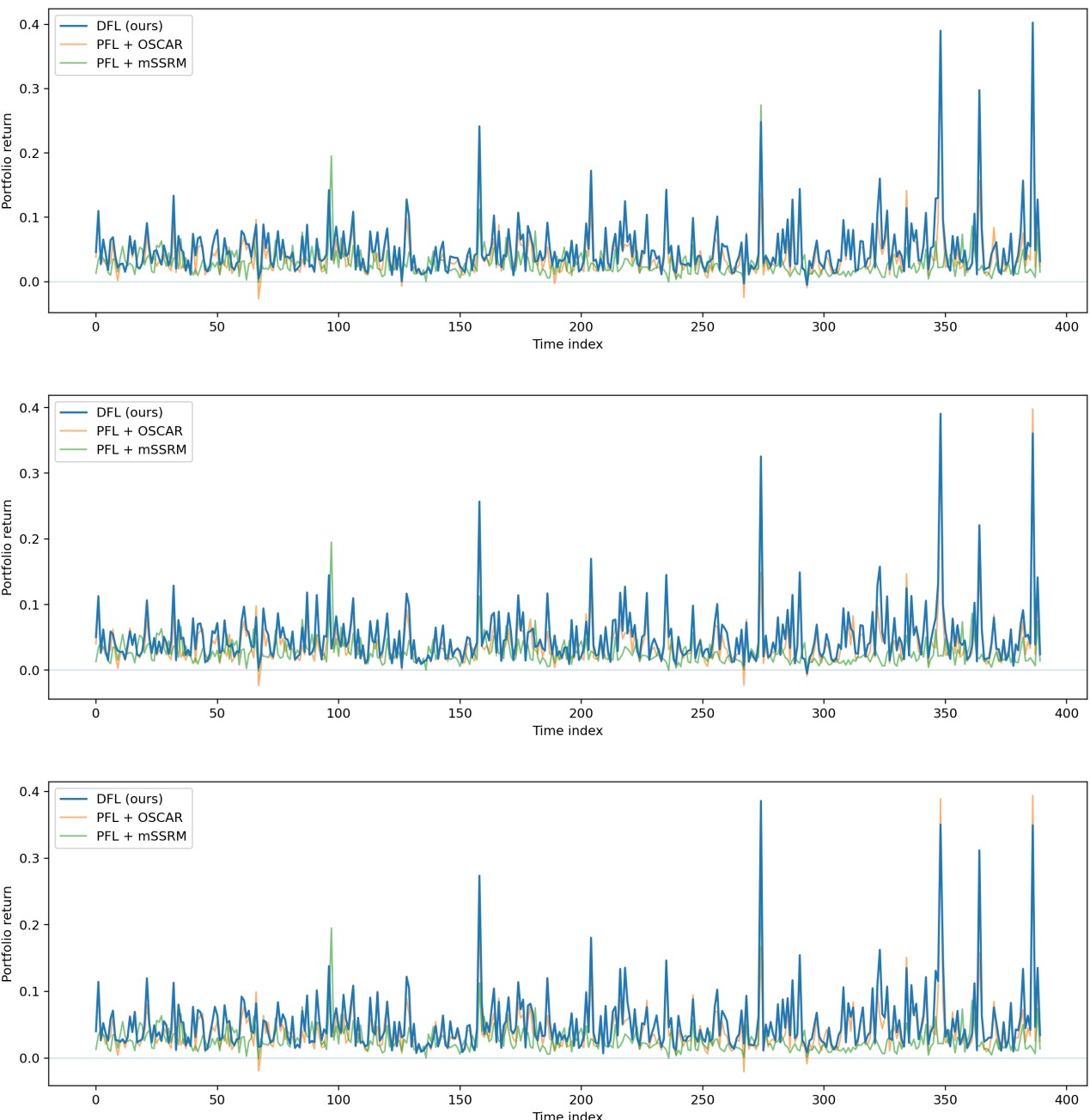

*Figure 3.* Portfolio return time-series of EuroStoxx50 on the test set for $\rho = 10\%$, $15\%$, and $20\%$. We report our DFL framework and PFL baselines: OSCAR and mSSRM-PGA.

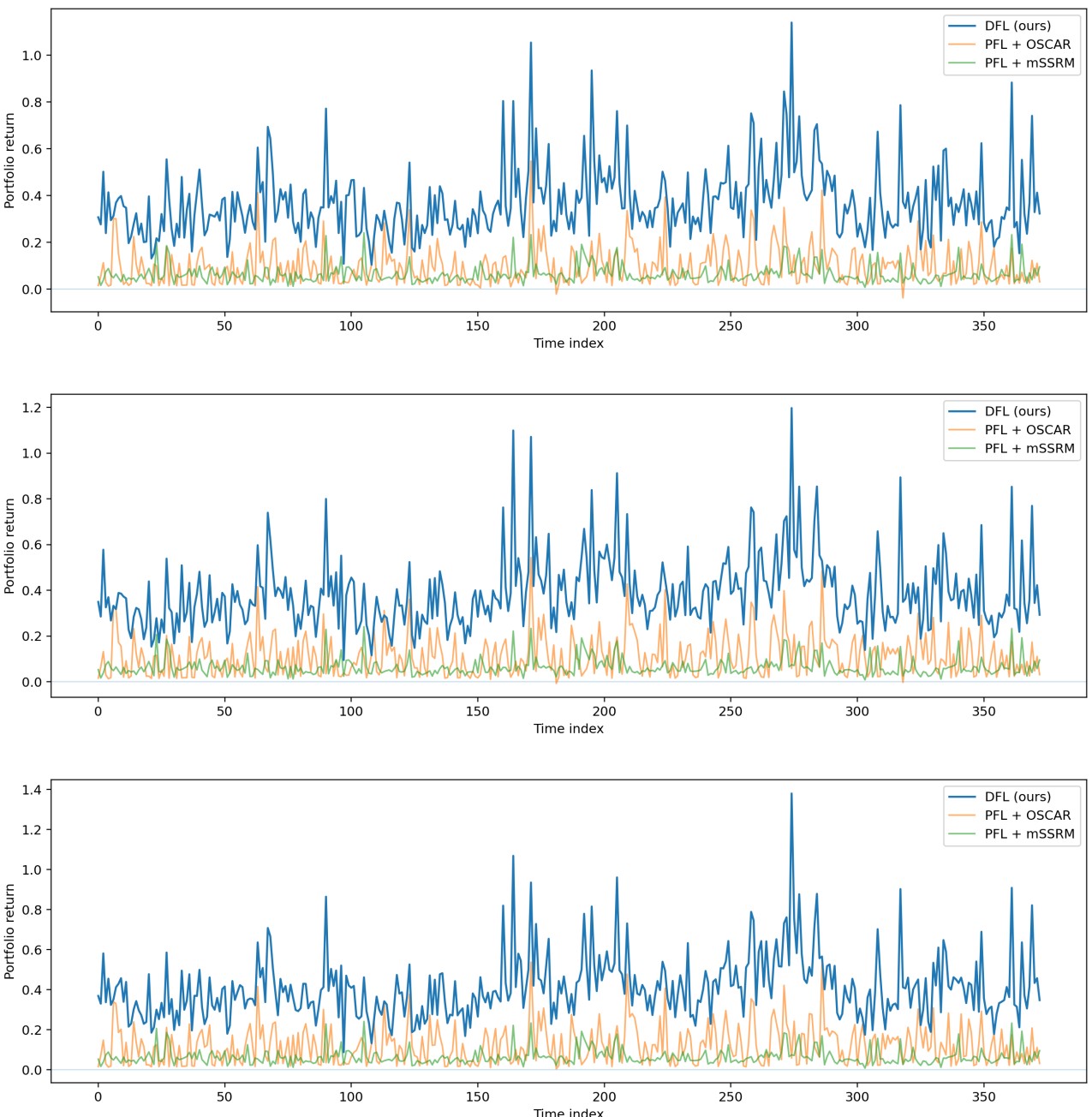

*Figure 4.* Portfolio return time-series of KOSPI200 on the test set for $\rho = 10\%$, $15\%$, and $20\%$. We report our DFL framework and PFL baselines: OSCAR and mSSRM-PGA.

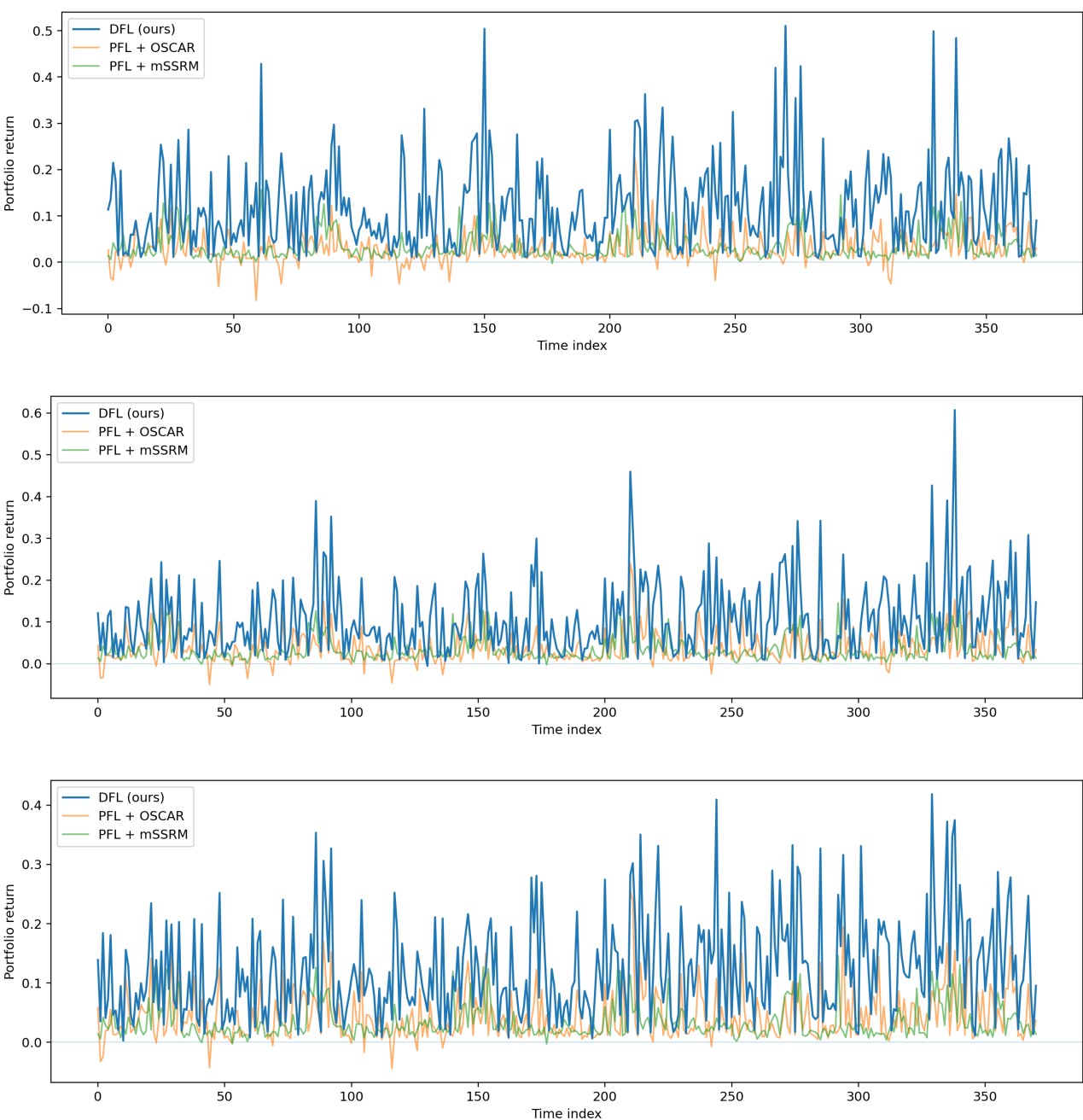

*Figure 5.* Portfolio return time-series of Nikkei225 on the test set for $\rho = 10\%$, $15\%$, and $20\%$. We report our DFL framework and PFL baselines: OSCAR and mSSRM-PGA.

