# OpenReview forum: "Decision-focused Sparse Tangent Portfolio Optimization"
_ICML.cc/2026/Conference — ICML 2026 regular_

### Official Review · Reviewer_7ct3 · 2026-02-25

**Soundness:** 2
**Presentation:** 2
**Significance:** 3
**Originality:** 3
**Overall Recommendation:** 4
**Confidence:** 4

**Summary:**

The paper proposes an end-to-end Decision-Focused Learning (DFL) framework for sparse tangent portfolio optimization. To address the NP-hard nature of cardinality-constrained optimization and the "predict-then-optimize" mismatch, the authors introduce two key technical innovations:


A DPP-compliant convex layer: They reformulate the non-convex Sharpe ratio maximization into a Disciplined Parametrized Programming compliant QCQP using homogenization and Cholesky factorization.


A differentiable "sum-to-k" operator: They replace discrete asset selection with a smooth, monotone squashing function that enforces an exact sparsity budget while allowing gradient flow.

The method is evaluated across four international equity markets (EuroStoxx50, FTSE100, KOSPI200, Nikkei225), demonstrating superior out-of-sample Sharpe ratios compared to traditional prediction-focused learning (PFL) and historical baselines.

**Compliance With Llm Reviewing Policy:**

Affirmed.

**Final Justification:**

Although this work has several experimental limitations, the results for the proposed method are still reasonably trustworthy.

**Key Questions For Authors:**

1. **Short-Selling & Margin**: Could you provide an empirical analysis of the short positions generated? Specifically, do these portfolios remain feasible when margin requirements are enforced?
2. **Sensitivity to $k$**: Since $k$ is a training parameter, how sensitive is the model to a mismatch between the $k$ used in training and the $k$ used in inference (e.g., training on $k=10\%$ but testing on $k=20\%$)? I appreciate highlighting the weaker results on EuroStoxx50. Could you further analyze whether the reduced performance of DFL and mSSRM is due to the smaller asset pool? More importantly, does the effectiveness of your method depend on the size of the candidate pool?
3. **Efficiency**: Please report the **elapsed time (wall-clock time)** for a standard training epoch compared to PFL baselines. How does this scale as the candidate asset pool size increases?
4. **Risk Metrics**: Beyond the Sharpe ratio, how does the model perform in terms of **Maximum Drawdown (MDD)**? Decision-focused models can sometimes chase returns at the expense of extreme tail risk.
5. **US Market Robustness**: Why were US large-cap markets excluded? Testing on the S&P 500 through 2025 would significantly strengthen the claim of robustness.

**Limitations:**

The authors discuss computational overhead and the restriction to DPP-compliant risk models. However, they should further address:

- The impact of asset pool size: Does the method's effectiveness diminish in significantly smaller pools like EuroStoxx50?

- The use of fixed asset counts vs. percentages: Why use percentages instead of a fixed number of assets (e.g., $k=10, 20$)?

**Strengths And Weaknesses:**

### **Strengths**

- **Originality**: The work provides a novel combination of **differentiable convex optimization layers** and a **smooth top-k operator** specifically tailored for the tangency direction of the mean-variance frontier.

- **Technical Soundness**: The derivation of the Jacobian for the top-k operator and the use of the implicit function theorem for the optimization layer are mathematically rigorous.

- **Significance**: Bridging the gap between forecasting and decision-making in sparse portfolios is a high-value problem in quantitative finance, especially for reducing transaction and monitoring costs.

- **Presentation**: The paper is well-structured, with clear algorithmic descriptions and a detailed explanation of the transition from non-convex to DPP-compliant formulations.

### **Weaknesses**

- **Restrictive Assumptions**: The model assumes **short selling is allowed** (unconstrained $w$). While this facilitates the mathematical transformation, it ignores real-world margin requirements and regulatory constraints, which could lead to "unreasonable" short positions in practice.

- **Generalizability and Efficiency**: The sparse budget $k$ is a fixed input to the training loop. This implies that a change in the desired number of assets requires a full re-training of the predictive model, limiting its flexibility. Furthermore, there is no report on the **wall-clock time (elapsed time)** compared to PFL baselines, which is critical given the overhead of differentiating through solvers.

- **Evaluation Breadth**:

  - **Market Bias**: The datasets focus on medium-cap or geographically specific markets (Europe, Japan, Korea). The lack of US large-cap markets (e.g., S&P 500 or NASDAQ 100) is a notable omission, as these markets have different liquidity and volatility profiles.

  -  **Timeline**: The testing period ends in **2023**. Evaluating the model through **2024-2025** would better demonstrate robustness against recent "black-swan" events and high-volatility regimes.

-  **Missing Metrics and Baselines**: The results focus heavily on the Sharpe ratio. Analysis of tail risk (e.g., **Maximum Drawdown (MDD)**) is missing. Additionally, the paper does not compare against the recently published **ASMCVaR***, which also addresses autonomous sparse optimization but focuses on Conditional Value at Risk (CVaR).


Yizun Lin, Yangyu Zhang, Zhao-Rong Lai*, Cheng Li, " Autonomous Sparse Mean-CVaR Portfolio Optimization", the 41st International Conference on Machine Learning (ICML, main track), 2024.

---

> ### Author Rebuttal · Authors · 2026-03-31
>
> (W1) Thank you for this comment. We agree that allowing short selling makes the formulation less directly aligned with practical portfolio constraints such as margin requirements and regulatory limits, and we also agree that the empirical behavior of the resulting short positions is an important question.
>
> In the current paper, the short-selling-allowed setting is adopted primarily for methodological consistency with the underlying OSCAR-based sparse Sharpe-ratio framework. In particular, the structural propositions that motivate the optimize–select–re-optimize pipeline are derived under the short-selling-allowed formulation, and our goal here is to build a differentiable decision-focused formulation on top of that problem class rather than to redesign the portfolio constraints themselves. In this sense, the present paper should be viewed as a methodological study for sparse Sharpe-ratio optimization under the standard OSCAR setting.
>
> Regarding the empirical short positions specifically, we agree that this deserves further analysis. However, the current paper does not include an explicit margin model or leverage constraint, so we do not claim that the generated portfolios are directly feasible under practical margin requirements. A proper answer to that question would require introducing an additional margin- or leverage-constrained portfolio formulation and then reevaluating the resulting allocations under those constraints.  More broadly, we view long-only, leverage-limited, or margin-constrained extensions as important future directions.
>
> (W2) In our formulation, k is treated as a fixed task parameter, and each model is trained and evaluated for the same target cardinality level. This reflects the problem setting studied in the paper—learning a predictor specialized to a given sparsity budget—rather than an overlooked implementation detail. We will clarify this fixed-k scope more explicitly.
>
> We also agree that wall-clock efficiency is an important practical consideration. The current paper does not report elapsed-time comparisons against PFL baselines, and we acknowledge this as a limitation. Runtime is not a claimed advantage of our method; in fact, differentiating through the optimization layer together with the soft top-k step makes training heavier than simpler PFL pipelines. We therefore view both computational cost and cross-k flexibility as limitations of the current framework, while the main contribution of the paper is methodological: showing that a differentiable end-to-end formulation can improve sparse portfolio quality under a fixed cardinality budget.
>
> (W3) Regarding market coverage, we evaluate the method on four equity universes to test it across multiple markets while keeping the optimization pipeline computationally manageable. We did not include larger US universes such as the S&P 500 because the current DFL framework becomes substantially heavier as the asset universe grows. We agree that this limits evaluation breadth and will state this more clearly as a limitation.
>
> Regarding timeline coverage, we follow the same 2014--2023 sample period as the underlying OSCAR-style setup to keep the comparison within a consistent experimental regime. We agree, however, that extending the evaluation to 2024--2025 would provide a stronger robustness test under more recent high-volatility conditions, and we will note this as an important direction for future work.
>
> (W4) Regarding evaluation metrics, our paper focuses on sparse Sharpe ratio maximization, so we center the evaluation on out-of-sample Sharpe ratio to match the downstream objective. We agree, however, that additional tail-risk metrics such as MDD would provide a more complete view of portfolio behavior, and we will state this more clearly as a limitation.
>
> Regarding ASMCVaR, while it is a relevant recent work in sparse portfolio optimization, it targets a different downstream objective, namely CVaR rather than Sharpe-ratio maximization. For this reason, we prioritized baselines that address the same sparse Sharpe-ratio setting more directly.
>
> (Q1) Please refer to W1 above!
>
> (Q2) Please refer to W2 above!
>
> (Q3) Please refer to W3 above!
>
> (Q4) Please refer to W4 above!
>
> (Q5) Please refer to W3 above!

---

> > ### Author Rebuttal · Reviewer_7ct3 · 2026-04-01
> >
> > Thanks for your reply and clarification. I understand that the current paper is mainly positioned as a methodological study built on the standard OSCAR-based sparse Sharpe-ratio setting, and that the short-selling-allowed formulation is inherited for theoretical and methodological consistency.
> >
> > That said, for W1, if possible, I still encourage the authors to provide some simple statistical analysis of the portfolio weights in the appendix or discussion section. For example, summary statistics on the distribution of weights, the frequency and magnitude of short positions, or the overall concentration pattern would already be helpful. Even if the current formulation does not include explicit margin or leverage constraints, such statistics would give readers a clearer picture of the empirical behavior of the learned portfolios and help assess how practically aggressive the allocations are.
> >
> > For W4, I also understand why out-of-sample Sharpe ratio is treated as the primary metric, since it is aligned with the optimization objective. However, I would still like to emphasize that risk level remains important in portfolio evaluation even under an SR-maximization setting. Portfolios with similar SR can still differ substantially in tail-risk behavior and downside exposure. Therefore, if possible, it would be useful to report metrics such as MDD or CVaR (if available) in the appendix for reference, so that readers can better understand the practical risk profile of the proposed method.

---

> > > ### Author Response · Authors · 2026-04-02
> > >
> > > Thank you for the helpful follow-up. We agree that, even under the current short-selling-allowed formulation, it is useful to provide additional empirical diagnostics so that readers can better understand the practical behavior of the learned portfolios.
> > >
> > > To address this, we computed simple summary statistics of the portfolio weights, focusing on the average short ratio and gross exposure for DFL and PFL(OSCAR based). As below, DFL appears more aggressive than PFL in terms of both short exposure and total leverage:
> > >
> > > | Metric | DFL | PFL |
> > > |---|---:|---:|
> > > | short ratio | 0.442 | 0.096 |
> > > | gross exposure | 5.187 | 2.160 |
> > >
> > > We also computed Maximum Drawdown (MDD) as an additional downside-risk metric. In the example of same EuroStoxx50 setting, DFL shows noticeably larger drawdowns than PFL:
> > >
> > > |  | DFL | PFL |
> > > |---|---:|---:|
> > > | 10% | 0.531 | 0.232 |
> > > | 15% | 0.614 | 0.288 |
> > > | 20% | 0.575 | 0.264 |
> > >
> > > Taken together, these results suggest that the current short-selling-allowed formulation can indeed produce more practically aggressive portfolios, especially under DFL. We therefore view this as an important limitation of the current study. In the revision, we will add these summary statistics in the appendix and clarify more explicitly that our present contribution is primarily methodological: showing that decision-focused learning can be successfully applied to a cardinality-constrained sparse Sharpe ratio optimization problem. Extensions that incorporate more practical constraints, such as long-only, leverage-limited, or margin-constrained formulations, remain important future work.
> > >
> > > If there are any points in our rebuttal that you found insufficiently convincing, or aspects of the work that remain unclear, we are always happy to discuss further! Please do not hesitate to edit your review (we'll monitor them 24/7), or convey them through the AC/SAC. It would be our pleasure to address them in time.

---

### Official Review · Reviewer_S8jY · 2026-02-25

**Soundness:** 3
**Presentation:** 3
**Significance:** 3
**Originality:** 3
**Overall Recommendation:** 4
**Confidence:** 5

**Summary:**

The paper addresses the challenge of sparse tangent portfolio optimization, which seeks to maximize the Sharpe ratio while limiting the number of assets in a portfolio (cardinality constraint). Traditional methods may suffer from a predict-then-optimize mismatch, where predictive models are trained on statistical losses (e.g., MSE) that may not align with the final portfolio performance. The authors propose an end-to-end Decision-focused Learning (DFL) framework. It converts the non-convex Sharpe ratio maximization into a Disciplined Parametrized Programming (DPP)-compliant convex program by homogenization and Cholesky factorization. Besides, it includes a smooth selection layer that enforces a strict sum-to-k budget, which allows gradients to flow through the discrete asset selection step. Then it trains the model by a combined loss of decision regret and forecasting error to optimize directly for out-of-sample Sharpe ratios.

**Compliance With Llm Reviewing Policy:**

Affirmed.

**Key Questions For Authors:**

See weaknesses.

**Limitations:**

See weaknesses.

**Strengths And Weaknesses:**

Strengths

•	The use of Cholesky factorization to achieve DPP-compliance for the covariance matrix in the optimization layer is a skillful solution to a common hurdle in DFL for finance.

•	The sum-to-k constraint in the soft top-k operator may be better than standard softmax-based selection as it preserves the exact sparsity budget throughout training.

•	Testing on four diverse markets with a long (10-year) look-back period provides robust evidence of the effectiveness of the model in real-world conditions.

•	The select-and-reoptimize architecture is logically presented and addresses the cascading error problem in traditional pipelines.

Weaknesses

•	The performance relies on the convex combination of DFL and MSE loss ($\alpha$). It needs a sensitivity analysis to show how varying $\alpha$ affects the stability of gradients and final Sharpe ratios.

•	The requirement to solve two optimization problems per training step (Algorithm 1) may be computationally intensive for large asset universes. Hence it is better to provide a wall-clock time comparison against the baselines to see how DFL balances between investing performance and runtime.

•	It is better to provide more recent references for the readers to learn about this area, such as:

Yizun Lin, Yangyu Zhang, Zhao-Rong Lai, Cheng Li, Autonomous Sparse Mean-CVaR Portfolio Optimization, ICML 2024.

Zhao-Rong Lai, Haisheng Yang, Linear Trading Position with Sparse Spectrum, IJCAI 2025.

---

> ### Author Rebuttal · Authors · 2026-03-31
>
> (W1) Thank you for this comment. In the current paper, α is used as a fixed design parameter rather than as a hyperparameter that we optimize through a full sensitivity study. Our main goal was to test the proposed end-to-end sparse DFL pipeline under a single stable training configuration, not to identify the best mixing coefficient. We chose α=0.5 as a balanced default because it avoids collapsing to either extreme: α=0 corresponds to pure MSE/PFL training, whereas α=1 corresponds to pure decision-focused training.
>
> The reason for using this mixed objective is practical. In our setting, the training signal from the decision-focused term alone can become weak or unstable due to the combination of differentiable optimization and soft sparse selection, while the MSE term provides a direct predictive signal that helps stabilize learning. We agree, however, that the current paper does not include a full α-sensitivity analysis for gradient stability or final Sharpe ratios. We will make this limitation explicit and clarify that the reported gains should be understood as results under this fixed mixed-loss setup, rather than evidence that α=0.5 is an optimized or universally best choice.
>
> (W2) We agree that the need to solve two optimization problems per training step makes the current DFL framework computationally heavier than the simpler PFL baselines, especially as the asset universe grows. We did not emphasize wall-clock efficiency in the current paper because computational cost is not a claimed advantage of our method. Our main contribution is methodological: showing that an end-to-end differentiable formulation can improve sparse portfolio quality under cardinality constraints. In contrast, runtime and scalability are limitations of the current framework rather than strengths. In particular, for larger asset universes, the repeated optimization-layer solves together with the soft top-k step can become a significant training bottleneck.
>
> We agree that the need to solve two optimization problems per training step makes the current DFL framework computationally heavier than the simpler PFL baselines, especially as the asset universe grows. In practice, we also observed a substantial increase in computational cost when running experiments on larger asset universes. We therefore regard scalability and runtime as limitations of the current framework rather than strengths. Our main contribution is methodological: showing that an end-to-end differentiable formulation can improve sparse portfolio quality under cardinality constraints.
>
> (W3) At the same time, our literature review was organized around methods that are most directly comparable to the problem studied in this paper, namely cardinality-constrained Sharpe ratio maximization. While the suggested papers also consider sparse portfolio optimization, they do not target the same objective as ours. In particular, our focus is on the sparse Sharpe ratio maximization problem, which is especially challenging due to the combination of a ratio-based objective and explicit sparsity constraints, and our methodological contribution is to make this class of problems amenable to end-to-end decision-focused learning. For this reason, we prioritized references and baselines that directly address sparse Sharpe-ratio optimization.

---

> > ### Author Rebuttal · Reviewer_S8jY · 2026-04-03
> >
> > The authors have fully addressed my concerns. After reading all the comments from all the reviewers, I would like to keep my score as it is already positive.

---

> > > ### Author Response · Authors · 2026-04-06
> > >
> > > Thank you very much for the thoughtful follow-up and for taking the time to read our rebuttal carefully. We sincerely appreciate that you found our responses adequate and that your concerns were fully addressed.
> > >
> > > We are especially grateful that you recognized several core strengths of the paper: the Cholesky-based route to DPP compliance in the optimization layer, the exact-budget soft top-k operator that preserves the cardinality constraint during training, and the select-and-reoptimize architecture that directly addresses the cascading mismatch of traditional predict-then-optimize pipelines. We also appreciate your recognition that the experiments over four diverse markets and a long evaluation horizon provide meaningful evidence of practical effectiveness.
> > >
> > > For the camera-ready version, we will make the paper more explicit about the current method’s main limitations and scope. Following related discussions with other reviewers, we will clarify in the revision why the mixed-loss weight alpha was fixed in the current study and add a brief sensitivity analysis in appendix. This will help explain the role of alpha as a practical stabilization parameter and illustrate its effect on training stability. We will also expand the discussion of runtime and scalability, making clearer that the repeated optimization steps required during training introduce substantial computational overhead relative to simpler PFL baselines, particularly for larger asset universes. In addition, we will refine the related-work section to better situate our contribution with respect to recent sparse portfolio optimization studies while clarifying the distinct focus of our paper on end-to-end decision-focused learning for cardinality-constrained sparse Sharpe ratio optimization.
> > >
> > > More broadly, we will sharpen the limitations discussion by directly acknowledging the current framework’s reliance on the mixed objective for stable training, its computational cost, and its limited scalability in larger universes. We hope these revisions will make the final version clearer for readers and more useful for assessing the paper’s contribution in the appropriate scope.

---

### Official Review · Reviewer_ao9E · 2026-03-12

**Soundness:** 2
**Presentation:** 2
**Significance:** 3
**Originality:** 3
**Overall Recommendation:** 4
**Confidence:** 4

**Summary:**

This paper considers the  cardinality-constrained Sharpe-ratio portfolio optimization. The proposed algorithm embedds the OSCAR algorithm into a decision-focused learning framework. The framework of the proposed method is clear: it follows the pipeline: ``score $\to$ differentiable top-$k$ selection $\to$ re-optimization'' structure, where the key implementation step is to replace the discrete selection stage with a soft top-$k$ operator so that gradients can flow through the pipeline. The paper further combines the decision-focused loss with an MSE term to stabilize training.

**Compliance With Llm Reviewing Policy:**

Affirmed.

**Final Justification:**

Based on the authors' rebuttal, the proposed framework is empirically usefull. I would raise my score.

**Key Questions For Authors:**

1.On the EuroStoxx50 dataset, the proposed method does not outperform PFL method. Could the authors provide further analysis explaining this behavior?

2.The paper introduces a new prediction loss to address potential gradient issues in the original $L_{\text{DFL}}$ objective, but provides limited theoretical justification. Could the authors clarify under what conditions the gradient problem arises and provide ablation studies showing the effect of this modification?

**Limitations:**

See weaknesses

**Strengths And Weaknesses:**

Strengths:

The main idea is clear and easy to follow. In particular, the paper gives a coherent differentiable version of the OSCAR-style pipeline: first solve the unconstrained tangency portfolio, then compute Cholesky-based scores, apply a soft top-$k$ selection rule, and finally re-optimize on the selected support. The empirical results suggest that this framework can work reasonably well in practice.



Weaknesses:
  1.  The use of differentiable or soft top-$k$ operators is standard in many learning and optimization settings.  The main novelty seems to lie more in integrating existing DDP within this specific sparse Sharpe-ratio portfolio problem than in proposing a new optimization technique.

2.    The paper should revise the notation $w$. In Section 3.1, $w$ denotes the portfolio weight vector, whereas in Section 3.2 the generic decision-focused learning formulation uses $x^\star(w)$, where $w$ instead plays the role of the problem parameter. This creates unnecessary confusion.


2.    The paper states that  decision-focused loss may suffer from noisy or diminishing gradients, and therefore introduces a convex combination $ L_{\text{Task}} = \alpha L_{\text{DFL}} + (1-\alpha)L_{\text{MSE}}$. However, the paper does not provide a clear ablation or sensitivity analysis over $\alpha$, nor does it provide theoretical analysis explaining when and why this combination resolves the gradient issue. This makes it difficult to assess whether the improvement comes from the proposed decision-focused formulation itself or from the addition of a standard prediction loss.

---

> ### Author Rebuttal · Authors · 2026-03-31
>
> (W1) Thank you for this comment. We agree that differentiable or soft top-k operators are not new by themselves, and our contribution is not to introduce a universally new top-k technique. Rather, the main novelty of this paper is to show that decision-focused learning can be successfully applied to a sparse Sharpe ratio portfolio problem that appears difficult to handle end-to-end because of its nontrivial objective, cardinality constraint, and downstream optimization structure.
>
> In particular, our contribution is a problem-specific differentiable formulation that makes this complex sparse portfolio problem trainable within a DFL framework. By combining a DPP-compatible reformulation of Sharpe ratio maximization with soft support selection and reoptimization, we obtain a fully differentiable pipeline for prediction, selection, and portfolio construction. We will revise the paper to make this positioning clearer and avoid overstating the novelty of the soft top-k component itself.
>
>
> (W2) We agree that the notation is currently overloaded. In Section 3.1, w denotes the portfolio weight vector, whereas in Section 3.2 the generic DFL formulation uses x^*(w), where w instead refers to the problem parameter. We will revise Section 3.2 to use a different symbol for the generic problem parameter (e.g., c) so that w is reserved consistently for portfolio weights throughout the paper.
>
>
> (W3) First, we would like to clarify that we do not introduce a new prediction loss. The additional term in Eq. (14) is the standard MSE loss, used as an auxiliary training signal together with the decision-focused term. Our intention was not to propose a new loss function in isolation, but to use a simple mixed objective as a stable default training setup for the proposed end-to-end differentiable sparse decision pipeline.
>
> The motivation is that pure decision-focused training can yield weak or unstable gradients in our setting, especially because the optimization pipeline includes both a differentiable convex layer and a soft sparse-selection step. In such cases, the decision loss may provide gradients that are noisy, small in magnitude, or locally unstable, whereas the MSE term provides a direct predictive supervision signal. The combined objective was therefore used as a practical compromise to preserve task-level alignment while stabilizing optimization. We agree that this point was not explained clearly enough in the current draft.
>
> At the same time, we acknowledge that the paper does not currently include a full sensitivity analysis over alpha, nor a formal theory characterizing exactly when the convex combination resolves the gradient issue. In our experiments, alpha=0.5 was used as a fixed default rather than a validation-tuned optimum: alpha= reduces to pure PFL/MSE training, while alpha=1 reduces to pure DFL. Since our goal was to evaluate a non-degenerate setting that keeps both predictive supervision and decision-level alignment active, we chose the midpoint as a simple default.
>
> We agree, however, that additional ablations would strengthen the paper. We will clarify that the empirical gains should be interpreted as coming from the full training formulation rather than attributing them solely to the decision loss term in isolation. We will also make it explicit that a broader alpha-sensitivity study and a more formal analysis of the gradient-stabilization effect are important directions for future work.
>
>
> (Q1) We agree that EuroStoxx50 is the one market where our method does not outperform the strongest PFL baseline on average. We believe this reflects a relatively easier regime for prediction-focused learning: compared with the larger markets, the downstream sparse selection problem is less demanding, so a standard predictor can already provide sufficiently useful return estimates for the optimizer. This interpretation is consistent with the market-wise pattern already reported in Appendix B.2, where we explicitly note that EuroStoxx50 is the only case in which our method is slightly below the strongest baseline, whereas on KOSPI200 and Nikkei225 it consistently achieves higher Sharpe ratios and expected returns.
>
>
> (Q2) Please refer to W3 above!

---

> > ### Author Rebuttal · Reviewer_ao9E · 2026-04-02
> >
> > Thank you for the clarification. The rebuttal makes the role of the mixed objective clearer: the auxiliary term is the standard MSE loss and is used as a practical stabilization mechanism rather than as a novel methodological contribution. This addresses the presentation issue to some extent.
> >
> > However, my main concern is only partially resolved. The claim that the mixed objective mitigates weak or unstable gradients remains heuristic, since the rebuttal does not provide either (i) an ablation or sensitivity analysis over $\alpha$, or (ii) a more formal characterization of when the decision-focused gradient becomes weak and how the MSE term helps. As a result, it is still difficult to assess whether the reported gains should be attributed to the proposed decision-focused sparse pipeline itself, or to the addition of a standard supervised regularization term.

---

> > > ### Author Response · Authors · 2026-04-04
> > >
> > > We appreciate the reviewer’s follow-up comment. We agree that the role of the mixed objective should be supported more explicitly, and that the current version does not yet provide a full sensitivity analysis over alpha.
> > >
> > > To better understand this issue, we conducted a small case study on EuroStoxx50 with k=20, using five different another random seeds and varying alpha in the mixed objective. The results suggest that the choice of alpha matters in practice:
> > >
> > > | alpha | SR |
> > > |---|---:|
> > > | 0   | 1.491 $\pm$ 0.185 |
> > > | 0.1 | 1.439 $\pm$ 0.151 |
> > > | 0.5 | 1.933 $\pm$ 0.117 |
> > > | 0.9 | 1.940 $\pm$ 0.594 |
> > > | 1.0 | 1.540 $\pm$ 0.397 |
> > >
> > > As shown above, very small values such as alpha = 0 or 0.1, where training is dominated by the MSE term, led to relatively lower Sharpe ratios. At the other extreme, alpha = 1, corresponding to pure decision-focused training, also underperformed the mixed setting and showed larger variability across seeds. By contrast, intermediate values such as alpha = 0.5 and 0.9 achieved the best average Sharpe ratios, although alpha = 0.9 exhibited a noticeably larger standard error than alpha = 0.5.
> > >
> > > These observations are consistent with our empirical motivation for using a mixed objective: a moderate supervised term can help stabilize training, while both an excessively dominant MSE term and a purely DFL objective can be suboptimal.
> > >
> > > As a view of the relationship between DFL and MSE work, a more formal intuition is as follows. Recent studies [1] have used Hessian eigenvalue analyses to characterize the geometry of DFL objectives, since the eigenvalue spectrum reflects the local curvature of the loss landscape and thus the quality of the gradient signal. Under this view, pure DFL tends to have eigenvalues concentrated near zero, indicating a largely flat decision-loss landscape. This means that local perturbations in prediction often induce only weak decision-aware gradients. A good survey from Mandi et al. [2] illustrates this in Section 2.2 with a knapsack example, where predictions with identical squared error can lead to entirely different decisions depending on whether the error crosses the decision boundary. As another simple example, consider a portfolio optimization with no-short constraints where the obj is to maximize the return, and two stocks A and B with true expected returns of 5.0% and 5.1%. The optimal allocation invests 100% in stock B. In this setting, a small prediction error predicting stock A and B as 5.0% and 4.9% will shift the entire allocation to stock A.
> > >
> > > In contrast, MSE provides a clearer optimization direction by directly penalizing prediction errors. When the two objectives are combined, the curvature becomes more balanced, yielding a gradient signal that is both more stable and still decision-aware. Therefore, the role of the MSE term is to improve optimization precisely in regimes where the pure decision-focused gradient is weak or poorly informative.
> > >
> > > At the same time, we agree with the reviewer that this evidence is still limited and should not be overstated. In the current submission, alpha was chosen empirically as a practical default rather than through a comprehensive sensitivity study. We will therefore revise the paper to make this point more precise and include an explicit sensitivity analysis over alpha in the appendix. More broadly, we also agree that understanding how prediction-focused guidance can most effectively assist DFL training remains an important direction for future work, rather than something fully resolved in the current paper.
> > >
> > > If there are any points in our rebuttal that you found insufficiently convincing, or aspects of the work that remain unclear, we are always happy to discuss further! Please do not hesitate to edit your review (we'll monitor them 24/7), or convey them through the AC/SAC. It would be our pleasure to address them in time.
> > >
> > > ```
> > > 1. Ghorbani, B., Krishnan, S., and Xiao, Y. An Investigation into Neural Net Optimization via Hessian Eigenvalue Density. 2019.
> > > 2. Mandi, J. et al. Decision-focused learning: Foundations, state of the art, benchmark and future opportunities. 2023.
> > > ```

---

### Official Review · Reviewer_wHME · 2026-03-12

**Soundness:** 3
**Presentation:** 3
**Significance:** 3
**Originality:** 3
**Overall Recommendation:** 4
**Confidence:** 4

**Summary:**

This paper addresses the problem of constructing sparse tangent portfolios -- portfolios that lie along the tangency direction of the mean-variance efficient frontier but are restricted to hold at most k assets. The standard cardinality-constrained Sharpe ratio maximization (Formulation 2, line 160) is NP-hard due to the discrete asset selection step, and conventional predict-then-optimize pipelines train forecasting models under statistical losses (e.g., MSE) that are misaligned with downstream portfolio performance. The authors propose an end-to-end decision-focused learning (DFL) framework that makes the entire sparse portfolio construction pipeline differentiable. Three technical ingredients are combined. First, the non-convex Sharpe ratio objective is reformulated via a classical homogenization trick (Iyengar and Kang, 2005) into a DPP-compliant convex quadratically constrained quadratic program (Eq. 5, line 243), enabling differentiation through the solver via CVXPY Layers. Second, a novel smooth top-k operator replaces the hard combinatorial asset selection: sigmoid functions are applied with a shared scalar shift found by bisection so that the soft mask sums to exactly k (Eq. 6, lines 264-266), and the authors derive a closed-form vector-Jacobian product (Eq. 11, line 303) that avoids forming the full Jacobian. Third, the pipeline follows a select-and-reoptimize structure inspired by the OSCAR heuristic (Bae et al., 2025), where Cholesky-space scores determine which k assets are selected before the Sharpe ratio is re-maximized on the restricted universe. The training loss is a convex combination of a regret-style DFL loss and a standard MSE prediction loss (Eq. 14, line 296), with the mixing parameter alpha fixed at 0.5 (Appendix A.2, line 563). Experiments on four equity index universes (EuroStoxx50, FTSE100, KOSPI200, Nikkei225) with sparsity levels k in {10%, 15%, 20%} show that the DFL framework consistently achieves higher out-of-sample daily Sharpe ratios than both historic baselines and PFL pipelines paired with three state-of-the-art sparse optimization back-ends (OSCAR, SD-relaxation, mSSRM-PGA). The paper also provides code at an anonymous repository (Appendix A.1).

**Compliance With Llm Reviewing Policy:**

Affirmed.

**Key Questions For Authors:**

(1) Lines 354-359 state that "the input covariance matrix Sigma is fixed based on the same historical period" and the dataset spans January 2014 to December 2023 with an 80/20 train/test split. Can the authors clarify whether the covariance matrix is estimated from the full 10-year sample (including test data) or only from the training portion? If the former, this constitutes look-ahead bias. How would results change if Sigma were estimated solely from training data, or updated with a rolling window? This directly affects the validity of the experimental conclusions.

(2) In Eq. 12 (line 300), the sharpness parameter beta is introduced to control how closely the soft top-k approximates the hard top-k. What value of beta is used in the experiments? Was any sensitivity analysis performed? If beta is too small the mask is overly smooth and the sparsity constraint is only loosely enforced; if beta is too large the gradients may vanish for most entries. Understanding this trade-off is important for practitioners and for assessing the reliability of the method.

(3) The combined loss in Eq. 14 (line 296) uses a fixed alpha=0.5 throughout all experiments (Appendix A.2, line 563). The authors state this was chosen "to balance prediction accuracy and decision quality." Was alpha tuned on a validation set, or was 0.5 simply a default? How sensitive are the Sharpe ratio results to this hyperparameter? Given that the DFL loss (Eq. 13) and the MSE loss operate on different scales, the effective weighting may not be what the nominal alpha suggests.

(4) Lines 414-418 note that "across all markets, the overlap between the assets selected by the two approaches is generally low." Can the authors quantify this overlap (e.g., Jaccard similarity averaged across time steps)? Additionally, is there a pattern in which types of assets DFL selects versus PFL -- for example, does DFL favor assets with lower individual volatility, or assets that provide better diversification within the k-asset subset?

(5) The paper does not discuss the computational cost of the DFL framework relative to the PFL baselines. Each DFL training iteration requires solving two QCQP instances per sample (before and after selection) plus the bisection for the soft top-k. Can the authors report wall-clock training times for DFL versus PFL, and discuss how these scale with n (number of assets) and k (sparsity budget)?

**Limitations:**

The authors discuss limitations in the concluding paragraphs of Section 7 (lines 441-452), identifying three: (i) computational overhead of the soft top-k operator for large asset universes, (ii) the approach being tailored to Sharpe ratio maximization and not easily extensible to other portfolio objectives, and (iii) the DPP-compliant parametrization restricting how risk information enters the optimization. These are honest and relevant. However, the limitation most likely to affect the validity of the reported results -- the fixed, potentially look-ahead-biased covariance matrix -- is mentioned only indirectly (lines 354-359) and deserves more explicit treatment. The absence of transaction cost analysis is another limitation that should be acknowledged given the paper's own framing around practical portfolio management.

**Strengths And Weaknesses:**

Strengths

(1) The central technical contribution -- making the cardinality-constrained portfolio optimization problem end-to-end differentiable -- is well motivated and cleanly executed. The three-stage differentiable decision layer (optimize, soft top-k select, re-optimize) shown in Figure 1 (lines 165-183) is an elegant architecture that addresses a genuine gap: prior DFL work largely avoids hard combinatorial constraints, and prior sparse portfolio work ignores end-to-end alignment of the predictor with the downstream objective. The smooth top-k operator in particular is a neat construction: the exact sum-to-k property (Eq. 6) avoids the need for projection or penalty tuning that plagues relaxation-based alternatives, and the closed-form vector-Jacobian product (Eq. 11) keeps backward-pass cost linear in n.

(2) The experimental comparison is well structured along two orthogonal axes: (i) how returns are modeled (Historic vs. PFL vs. DFL) and (ii) how the cardinality-constrained optimization is solved (OSCAR, SD-relaxation, mSSRM-PGA). This factorial design, described in Section 5.3 (lines 363-384), makes it straightforward to attribute performance gains to the DFL training paradigm rather than to the choice of optimization back-end. The DFL framework achieves the best Sharpe ratio in 11 out of 12 market-times-sparsity combinations in Table 1, with the sole exception being EuroStoxx50 at k=10% where PFL+OSCAR is slightly better (3.707 vs. 1.818). This breadth across four geographically diverse markets and three sparsity levels provides reasonable evidence of robustness.

(3) The reformulation of the Sharpe ratio objective from a non-convex fractional program into a DPP-compliant QCQP (Section 4.1, lines 220-248) is technically sound and clearly presented. The use of the Cholesky factorization to satisfy DPP's affine-parameter requirement (lines 234-240) is a practical insight that others working on DFL for portfolio problems could directly reuse.

(4) The paper provides an anonymous code link (Appendix A.1, line 555), specifies all hyperparameters (learning rate 0.001, batch size 64, alpha=0.5, bisection iterations 32, two hidden layers of 256 and 128 neurons -- Appendix A.2, lines 558-566), and reports standard errors over the full training set for all entries in Table 1. These details support reproducibility.

(5) The discussion of asset selection differences between PFL and DFL (lines 414-421) provides useful qualitative insight: the overlap in selected assets is generally low, suggesting that decision-focused training reshapes not just the portfolio weights but the composition of the selected asset set itself, a finding that goes beyond what Sharpe ratio numbers alone convey.

Weaknesses

Fundamental concerns:

(1) The covariance matrix is fixed and estimated once from the full sample period (lines 354-359: "we treat the cross-asset covariance structure as exogenous and time-invariant, estimated once and then held constant during training and evaluation"). This is a significant limitation for a paper targeting non-stationary financial markets. The Sharpe ratio depends jointly on mu and Sigma (Eq. 1), so learning mu end-to-end while treating Sigma as a known constant sidesteps half of the estimation problem. Moreover, the use of a full-sample covariance estimate introduces look-ahead bias: the covariance is computed using data that includes the test period. The authors acknowledge this limitation in Section 7 (lines 441-448) but do not quantify its impact. Without an ablation that, say, uses only training-period data for covariance estimation or compares rolling-window covariance estimates, it is unclear how much of the reported performance advantage is attributable to this favorable information leakage versus the DFL training itself.

(2) The experimental evaluation, while broad in market coverage, is narrow in several other dimensions. Only a single predictive model architecture is tested (a two-layer MLP with a 100-day lookback window, Appendix A.2). The paper does not explore whether the DFL advantage holds with stronger base predictors (e.g., LSTMs, transformers, or even linear models), which would be important for understanding when end-to-end training provides the most value. Additionally, no transaction cost analysis is reported despite the paper's own motivation that sparse portfolios reduce transaction costs (lines 011-012, 047-048). The daily rebalancing implied by the experimental setup (one-step-ahead predictions at every trading day) would incur substantial turnover, and the Sharpe ratio improvements could be eroded or reversed after costs.

(3) The paper lacks any formal analysis of the approximation quality of the soft top-k operator relative to the hard top-k. While the sharpness parameter beta is introduced in Eq. 12 (line 300), there is no discussion of how beta is chosen, what value is used in practice, or how sensitive the results are to this choice. Since the soft mask is continuous in (0,1) rather than binary, the "selected" assets receive weights slightly below 1 and the "rejected" assets receive weights slightly above 0 during the forward pass. How this affects the quality of the selected subset and the re-optimization step is not analyzed.

Minor concerns:

(4) The writing in the introduction is somewhat repetitive. The predict-then-optimize mismatch is explained three times with slightly different phrasing (lines 055-063, the right column of page 1 starting at "Yet even with strong heuristics," and again at lines 110-119). Tightening this would free space for additional experiments or analysis.

(5) Proposition 3.2 (lines 128-137) and Proposition 3.3 (lines 186-193) are stated without proof in the main text. While these results build on Kim and Lee (2016), the paper would benefit from at least proof sketches in the appendix to be self-contained. As it stands, the reader must consult external references to verify these claims.

(6) The standard errors reported in Table 1 for the DFL method are substantially larger than those for the Historic baselines (which have zero standard error by construction since they use no learned model). For instance, on Nikkei225 at k=10%, the DFL result is 3.434 +/- 0.194, while PFL+OSCAR is 0.967 +/- 0.824. The large standard errors for PFL baselines, sometimes exceeding the mean (e.g., PFL+OSCAR on Nikkei225 at k=10%: 0.967 +/- 0.824), suggest high sensitivity to training randomness, yet the paper does not report how many random seeds were used or provide confidence intervals for the difference between methods.

(7) The paper considers only equity index universes of modest size (50 to 225 assets). The authors acknowledge scalability concerns with the soft top-k operator for larger universes (lines 441-443), but no experiments are conducted on universes of, say, 500 or 1000 assets, where the computational cost of the bisection search and the CVXPY solver calls within each training iteration would become more pressing.

---

> ### Author Rebuttal · Authors · 2026-03-31
>
> (W1) Thank you for raising this point. We clarify that our experiments do not use a full-sample covariance estimate: covariance is estimated using only the training data, so there is no look-ahead leakage from the test period. We will revise the paper to make this explicit. In the current implementation, covariance is fixed over the training period mainly to keep the setup simple and consistent. At the same time, we agree that this is a simplifying assumption, and that rolling covariance estimation would likely provide a more realistic treatment of time-varying risk. We will therefore present the fixed-covariance setting more clearly as a simplification of the current study.
>
> (W2) Regarding the predictive architecture, our goal is to evaluate the DFL framework itself rather than the strength of a particular forecaster. As in many prior DFL studies, we use a simple and controlled predictor so that the comparison isolates the effect of decision-aware end-to-end training rather than architectural variation. Accordingly, we adopt a standard two-layer MLP with a fixed 100-day lookback window and leave stronger predictors to future work. Regarding transaction costs, we agree that they matter in practice, but they are outside the main scope of this paper. Our experiments report gross out-of-sample performance, and all methods are evaluated under the same one-step-ahead prediction and re-optimization protocol, so the relative comparison remains fair.
> ```
> 1. Mandi, J., Kotary, J., Berden, S., Mulamba, M., Bucarey, V., Guns, T., and Fioretto, F. Decision-focused learning: Foundations, state of the art, benchmark and future opportunities. 2023.
> 2. Lee, J., Jeon, H., Bae, H., and Lee, Y. Return Prediction for Mean-Variance Portfolio Selection: How Decision-Focused Learning Shapes Forecasting Models. 2025.
> ```
> (W3) This is a very good point. In our implementation, the sharpness parameter for the soft top-k operator was fixed to β=5 in all experiments. Empirically, when β is set to a small value (roughly 1~3), the resulting gate becomes overly smooth, whereas for larger values (around 10+), the gradient flow can become weaker and less stable. To illustrate this trade-off, we will include a representative example from EuroStoxx50.
>
> | β | Off-support Mass | Gradient |
> |---|---:|---:|
> | 1  | 3.84  | 0.41 |
> | 5  | 1.92  | 0.27 |
> | 20 | 0.404 | 0.06 |
>
> Off-support Mass indicates approximation sharpness, and Gradient is the l2-norm of the loss gradient w.r.t. mu. We will also add a brief multi-seed sensitivity analysis for β in the appendix.
>
> (W4)(W5) We agree and will address both points in the revision by tightening the Introduction and adding brief appendix proof sketches for Propositions 3.2 and 3.3.
>
> (W6) We used five random seeds for all learned methods. The nonzero standard errors reflect variability from training randomness, such as initialization and mini-batch sampling. We agree that five seeds provide only a limited estimate of variability, and we will state and clarify this experimental protocol more clearly in the revision. The overall pattern in Table 1 remains consistent, with DFL generally outperforming learned baselines.
>
> (W7) We agree that scalability to larger universes is a limitation of the current framework. In our preliminary experiments, the computational cost increased substantially on larger universes such as the S&P500, which is why we left this as a limitation rather than claiming broader scalability. We will make this point more explicit in the camera-ready version.
>
> (Q1) Please refer to W1 above!
>
> (Q2) Please refer to W3 above!
>
> (Q3) We thank the reviewer for this point and refer to our response to other reviewers on the sensitivity of alpha.
>
> (Q4) We quantify overlap using average Jaccard similarity between DFL and PFL selections. As shown in the table below, the average overlap is consistently low across markets and cardinality levels, supporting our claim that the two methods often select different subsets. For the second question, we conducted additional analysis but found the results too qualitative for a clear claim, so we will remove that discussion and focus on the quantified overlap.
>
> | Market | 10% | 15% | 20% |
> |---|---:|---:|---:|
> | EuroStoxx50 | 0.000 | 0.167 | 0.111 |
> | FTSE100 | 0.059 | 0.037 | 0.118 |
> | KOSPI200 | 0.067 | 0.116 | 0.125 |
> | NIKKEI225 | 0.000 | 0.067 | 0.050 |
>
> (Q5) We did not highlight computational cost because it is not an advantage of our DFL framework over PFL. On the contrary, the additional optimization solves and soft top-k step make DFL more expensive during training, especially as n grows. We therefore regard scalability and training cost as limitations of the current method, while the main contribution of the paper is improved decision quality rather than computational efficiency.

---

> > ### Author Rebuttal · Reviewer_wHME · 2026-04-05
> >
> > Thanks for the responses. I will keep my score .

---

> > > ### Author Response · Authors · 2026-04-06
> > >
> > > Thank you very much for the thoughtful follow-up and for taking the time to read our rebuttal carefully. We sincerely appreciate that you found our responses adequate and that the main concerns were addressed.
> > >
> > > We are especially grateful that you recognized the paper’s main technical strength: an end-to-end differentiable framework for sparse tangency-portfolio construction, built from a DPP-compliant Sharpe ratio reformulation, a soft top-k operator that preserves the exact cardinality budget with an efficient closed-form backward pass, and a select-and-reoptimize design tailored to cardinality-constrained portfolio decisions.
> > >
> > > For the camera-ready version, we will revise the paper to explicitly acknowledge the weakness of fixing the covariance matrix over the full training period. Although this avoids look-ahead leakage from the test period, it remains a simplifying assumption and may fail to reflect time-varying market risk. To better address this issue, we will include this point more clearly in the limitations and discuss rolling-window covariance estimation as an important next step toward a more realistic experimental setting. In addition, we will add the beta sensitivity discussion for the soft top-k operator to clarify the trade-off between approximation sharpness and gradient stability. We will also report more clearly that all learned methods were run with five random seeds, and we will tighten the discussion of variability and standard errors in the experimental section. Finally, we will incorporate the average Jaccard-overlap analysis into the paper to quantitatively support the claim that DFL and PFL often produce materially different selected subsets.
> > >
> > > More broadly, we will sharpen the discussion of limitations by more directly highlighting the computational overhead of the current framework and the absence of transaction-cost modeling as practical limitations of the present study.

---

### Decision · Program_Chairs · 2026-04-30

**Decision:**

Accept (regular)

**Comment:**

This paper introduces an end-to-end decision-focused learning framework for sparse tangent portfolio optimization. The reviewers agreed the core technical contributions, specifically the smooth top-k operator and the Cholesky-based convex optimization layer, are mathematically sound and well-executed.

The submission remained borderline during the initial review phase due to practical concerns regarding real-world applicability. In the rebuttal and discussion period, the authors constructively addressed these issues and explicitly acknowledged the framework's limitations, notably the fixed covariance assumption, computational overhead on large asset pools, and the tendency to output aggressive portfolios with higher maximum drawdowns and shorting risks. Reviewers appreciated these clarifications and agreed the technical contribution merits publication.

The paper is recommended for acceptance. Given the borderline consensus, the authors are fully expected to honor their rebuttal commitments and integrate the promised modifications, additional analyses, and objective discussions of practical limitations into the camera-ready manuscript.